EMBO
Molecular Medicine

# A *POGLUT1* mutation causes a muscular dystrophy with reduced Notch signaling and satellite cell loss

Emilia Servián-Morilla[1,2,†], Hideyuki Takeuchi[3,†,‡], Tom V Lee[4,†], Jordi Clarimon[2,5], Fabiola Mavillard[2,6], Estela Area-Gómez[7], Eloy Rivas[8], Jose L Nieto-González[2,6], Maria C Rivero[2,6], Macarena Cabrera-Serrano[1,2], Leonardo Gómez-Sánchez[2,6], Jose A Martínez-López[2,6], Beatriz Estrada[9], Celedonio Márquez[1], Yolanda Morgado[10], Xavier Suárez-Calvet[11,12], Guillermo Pita[13], Anne Bigot[14], Eduard Gallardo[11,12], Rafael Fernández-Chacón[2,6], Michio Hirano[7], Robert S Haltiwanger[3,‡], Hamed Jafar-Nejad[4] & Carmen Paradas[1,2,7,*]

## Abstract

Skeletal muscle regeneration by muscle satellite cells is a physiological mechanism activated upon muscle damage and regulated by Notch signaling. In a family with autosomal recessive limb-girdle muscular dystrophy, we identified a missense mutation in *POGLUT1* (protein *O*-glucosyltransferase 1), an enzyme involved in Notch posttranslational modification and function. *In vitro* and *in vivo* experiments demonstrated that the mutation reduces *O*-glucosyltransferase activity on Notch and impairs muscle development. Muscles from patients revealed decreased Notch signaling, dramatic reduction in satellite cell pool and a muscle-specific α-dystroglycan hypoglycosylation not present in patients' fibroblasts. Primary myoblasts from patients showed slow proliferation, facilitated differentiation, and a decreased pool of quiescent PAX7⁺ cells. A robust rescue of the myogenesis was demonstrated by increasing Notch signaling. None of these alterations were found in muscles from secondary dystroglycanopathy patients. These data suggest that a key pathomechanism for this novel form of muscular dystrophy is Notch-dependent loss of satellite cells.

**Keywords** muscular dystrophy; Notch; *O*-glycosylation; POGLUT1; satellite cell

**Subject Categories** Development & Differentiation; Musculoskeletal System

## Introduction

Cell surface glycans are diverse in structure and function and play critical roles in many biological processes including infection, cancer, and development (2009). Broadly speaking, glycans affect the function of proteins to which they are linked by modulating their structure and/or by serving as a direct recognition signal for other proteins (Varki & Sharon, 2009). Glycans are involved in protein folding, stability, and trafficking and regulate the activity of important signaling pathways such as Notch, WNT, BMP, and Hedgehog (Christian, 2000; Yan & Lin, 2009; Jafar-Nejad *et al*, 2010; Takeuchi & Haltiwanger, 2014). Not surprisingly, aberrant glycosylation leads to a variety of human diseases, the list of which is growing (Freeze *et al*, 2014).

Protein glycosylation can directly regulate signaling events. A clear example of this is the regulation of the evolutionarily conserved Notch signaling pathway, which is an intercellular

1 Neuromuscular Disorders Unit, Department of Neurology, Instituto de Biomedicina de Sevilla, Hospital U. Virgen del Rocío/CSIC/Universidad de Sevilla, Sevilla, Spain
2 Centro de Investigación Biomédica en Red sobre Enfermedades Neurodegenerativas (CIBERNED), Madrid, Spain
3 Department of Biochemistry and Cell Biology, Stony Brook University, Stony Brook, NY, USA
4 Department of Molecular and Human Genetics, Baylor College of Medicine, Houston, TX, USA
5 Memory Unit, Department of Neurology and Sant Pau Biomedical Research Institute, Hospital de la Santa Creu i Sant Pau, Universitat Autònoma de Barcelona, Barcelona, Spain
6 Department of Medical Physiology and Biophysics, Instituto de Biomedicina de Sevilla, Hospital U. Virgen del Rocío/CSIC/Universidad de Sevilla, Sevilla, Spain
7 Department of Neurology, Columbia University Medical Center, New York, NY, USA
8 Department of Pathology, Instituto de Biomedicina de Sevilla, Hospital U. Virgen del Rocío/CSIC/Universidad de Sevilla, Sevilla, Spain
9 Centro Andaluz de Biología del Desarrollo (CABD), Universidad Pablo Olavide, Sevilla, Spain
10 Department of Neurology, Hospital U. Valme, Sevilla, Spain
11 Laboratori de Malalties Neuromusculars, Institut de Recerca de HSCSP, Universitat Autònoma de Barcelona (UAB), Barcelona, Spain
12 Centro de Investigación Biomédica en Red sobre Enfermedades Raras (CIBERER), Barcelona, Spain
13 Human Genotyping Unit-CeGen, Spanish National Cancer Research Centre, Madrid, Spain
14 UPMC Univ Paris 06, INSERM UMRS974, CNRS FRE3617, Center for Research in Myology, Sorbonne Universités, Paris, France
*Corresponding author. Tel: +34 955923045; Fax: +34 955013536; E-mail: cparadas@us.es
†These authors contributed equally to this work
‡Present address: Complex Carbohydrate Research Center, The University of Georgia, Athens, GA, USA

communication mechanism widely used during animal development (Hori *et al*, 2013). The extracellular domain of the Notch receptor is modified with *O*-fucose and *O*-glucose glycans (Moloney *et al*, 2000). The structures of these glycans change in a tissue-specific and developmentally regulated manner (Johnston *et al*, 1997), and altering Notch glycosylation dramatically affects its activity (Stanley & Okajima, 2010; Takeuchi & Haltiwanger, 2010). One of the cell types whose regulation by Notch signaling has been intensely studied in recent years is the muscle-specific adult stem cell called the satellite cell (SC) (Mourikis & Tajbakhsh, 2014). SCs reside between the basal lamina and sarcolemma of myofibers (Mauro, 1961) and are the primary contributors to skeletal muscle growth and repair (Collins *et al*, 2005). Despite continuous regeneration, pools of SC are maintained in healthy muscle. This is accomplished through asymmetric cell divisions of SC, which give rise both to self-renewing SC and to committed myogenic progenitors that differentiate (Kuang *et al*, 2007; Sacco *et al*, 2008). In this process, the Notch pathway is key for maintaining quiescence in SC and for homing of SC-derived myoblasts (Bjornson *et al*, 2012; Brohl *et al*, 2012). A recent report indicates that mice in which Notch signaling is specifically blocked in SCs exhibit a decrease in the number of SCs and histological features of muscular dystrophy even upon normal daily activity (Lin *et al*, 2013), suggesting that Notch-mediated maintenance of an active SC pool is essential for repairing muscle damage caused by regular activity and for maintaining healthy muscle. Muscular dystrophies are inherited disorders characterized by progressive weakness due to skeletal muscle degeneration, due to mutations in a growing list of responsible genes (Chandrasekharan & Martin, 2010; Rahimov & Kunkel, 2013). However, to date, no primary molecular defects in SC or Notch pathway components have been identified in human muscular dystrophies.

Mutations in several genes disrupt various aspects of the dystrophin-associated glycoprotein complex, which links the cytoskeleton to the extracellular matrix in muscle (Chandrasekharan & Martin, 2010). A key component of this complex is dystroglycan (Ervasti *et al*, 1990), a transmembrane protein essential for normal basement membrane development and muscle maintenance. Dystroglycan comprises a transmembrane β-subunit non-covalently linked to an extracellular α-subunit containing a mucin-like domain, which is modified with numerous *O*-linked glycans (Barresi & Campbell, 2006). The β-subunit is linked to the actin cytoskeleton, and the *O*-linked glycans on the α-subunit are critical for its binding capacity to extracellular matrix proteins such as laminin and agrin (Ervasti & Campbell, 1993; Michele *et al*, 2002). Mutations in dystroglycan itself result in the primary dystroglycanopathies (Henry & Campbell, 1998; Cote *et al*, 1999; Hara *et al*, 2011; Willer *et al*, 2014; Riemersma *et al*, 2015; Kanagawa *et al*, 2016). Secondary dystroglycanopathies are caused by interruption of α-dystroglycan–ligand interactions due to mutations in a growing list of genes [currently eighteen (Bonnemann *et al*, 2014)] encoding glycosyltransferases and accessory proteins responsible for α-dystroglycan's extensive posttranslational modifications (Muntoni *et al*, 2011; Inamori *et al*, 2012; Yoshida-Moriguchi *et al*, 2013). These include protein *O*-mannosyltransferase 1 (POMT1) and POMT2, which add *O*-linked mannose to α-dystroglycan (Manya *et al*, 2004; Willer *et al*, 2004), and like-acetylglucosaminyltransferase (LARGE), which is responsible for the addition of repeating xylose-glucuronic acid units on the *O*-mannosyl glycans

on α-dystroglycan (Inamori *et al*, 2012). The dystroglycan gene (*DAG1*) is expressed in SCs (Cohn *et al*, 2002), but glycosylated α-dystroglycan is not required by myogenic cells during proliferation (Awano *et al*, 2015), a notion supported by the observation that trace amounts of glycosylated α-dystroglycan are found in C2C12 and human primary myoblasts during proliferation. Cultures of freshly isolated muscle fibers suggest that glycosylation of α-dystroglycan is needed for proliferation of SCs *in vivo*, but it is not necessary when SCs are removed from their niche (Ross *et al*, 2012). The transition to larger glycans occurs rapidly after differentiation is induced, and a direct correlation exists between LARGE-dependent extension of *O*-glycans on α-dystroglycan and its function as an extracellular matrix receptor (Goddeeris *et al*, 2013). Despite extensive studies regarding the genetics of dystroglycanopathies, the responsible gene is still unknown in a significant number of patients; thus, mutations in additional genes will likely be implicated in this group of diseases.

Here, we report a homozygous missense mutation (D233E) in the protein *O*-glucosyltransferase 1 gene, *POGLUT1*, in four siblings with autosomal recessive limb-girdle muscular dystrophy. POGLUT1 is the sole enzyme that directly adds *O*-glucose to a distinct serine residue of epidermal growth factor-like (EGF) repeats containing a CXSX(P/A)C consensus sequence (Rana *et al*, 2011), many of which are found in Notch extracellular domains (Fernandez-Valdivia *et al*, 2011; Rana *et al*, 2011). The D233E mutation dramatically reduces the *O*-glucosyltransferase activity of POGLUT1, leading to impaired Notch signaling and a dramatic decrease in the number of SCs in adult muscles. Transgenic experiments in *Drosophila* indicate that D233E impairs the ability of human POGLUT1 to rescue the muscle phenotype caused by the loss of fly Poglut1 activity. Patient muscles show α-dystroglycan hypoglycosylation and decreased binding to laminin, but normal binding to agrin and normal basement membrane structure. Moreover, unlike other dystroglycanopathies, patient fibroblasts exhibit normal α-dystroglycan glycosylation and laminin binding. Together, our findings indicate that exhaustion of the SC pool plays a primary role in this novel form of muscular dystrophy.

## Results

### Clinical and radiological findings

A consanguineous family from southern Spain comprises 17 individuals spanning three generations (Fig 1A). Four out of five siblings from generation II presented a phenotype consistent with a limb-girdle muscular dystrophy. Specifically, the patients exhibited muscle weakness predominantly in the proximal lower limbs, with onset during the third decade. The disease course was progressive, leading to scapular winging and wheelchair confinement. For more extended clinical data regarding this family, see the Appendix Information, Appendix Fig S1, and Appendix Tables S1 and S2. Serum creatine kinase level was normal in three patients and mildly elevated in one (Appendix Table S1). Muscle biopsies from all four affected siblings revealed histological features ranging from very mild myopathic changes to classic dystrophic pathology (Fig 1A). Proteins typically affected in myopathies displayed normal expression in muscle, except for a reduction in α-dystroglycan

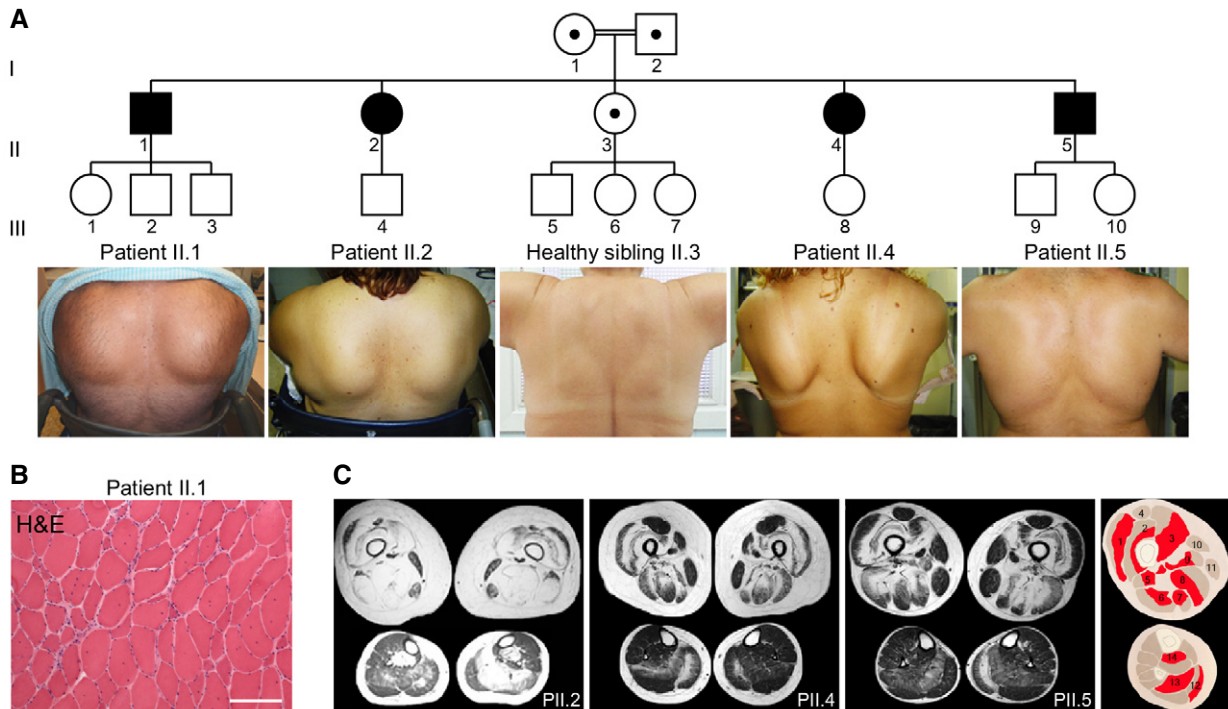

**Figure 1. POGLUT1 missense mutation in a family with a limb-girdle muscular dystrophy.**

A   The family pedigree, where circles denote female members, squares male members, solid symbols affected members, and white symbols asymptomatic members with normal physical exam; the dots indicate heterozygous carriers, and double line denotes a consanguineous marriage. The pictures show scapular winging, which is a consistent clinical sign in affected individuals.

B   Hematoxylin and eosin staining (H&E) of skeletal muscle from patient II.1 shows histological features of moderate-to-severe dystrophic pattern. Scale bar, 50 μm.

C   T1-weighted MRI axial images at thigh and calf levels show that the fatty degeneration is more prominent in thigh muscles, equally affecting posterior and anterior compartments, with relative sparing of the rectus femoris, sartorius, and gracilis muscles until late stages (4, 10, and 11, respectively). Strikingly, the fatty tissue is located in the internal parts of almost all the affected muscles in thigh (1, 2, 3, 5–9), while the external regions are spared. At calf level, only the gastrocnemius medialis muscle (12) shows this pattern, while the soleus (13) is diffusely involved. Patient II.2 (PII.2) shows late-stage thigh muscles with an unusual involvement of the tibialis posterior muscle (14) in the lower leg.

(Appendix Fig S2). Muscle magnetic resonance imaging (MRI) of the legs revealed a striking pattern of muscle involvement (Fig 1C), with early fatty replacement of internal regions of thigh muscles that spared external areas. This "from inside-to-outside" mode of fatty degeneration progressed over the years and did not match the distribution patterns typically associated with other forms of muscular dystrophies (Appendix Information and Appendix Figs S3 and S4).

**Expression and functional modification of α-dystroglycan in patients**

Given the key role played by aberrant α-dystroglycan glycosylation and function in a subset of muscular dystrophies and because of the observed decrease in α-dystroglycan levels in patient muscles, we examined the glycosylation status and ligand-binding ability of α-dystroglycan in our patients. Immunofluorescence staining of frozen cross sections from skeletal muscle biopsy with an antibody against glycosylated α-dystroglycan [IIH6 (Ervasti & Campbell, 1991)] revealed a variable reduction in the glycosylated form of α-dystroglycan at the sarcolemma in patients, while antibodies against α-dystroglycan core protein, β-dystroglycan, and laminin α2

showed normal staining (Fig 2A and Appendix Fig S5A). In agreement with this observation, Western blots showed a reduction in α-dystroglycan glycosylation in patient muscle, accompanied by a mild decrease in the molecular weight of glycosylated α-dystroglycan compared with controls. To examine whether decreased α-dystroglycan glycosylation affected binding to ligands, we performed a ligand overlay assay. As shown in Fig 2B, the laminin-binding activity was diminished in muscle. However, the agrin-binding activity to the patients' muscle extracts showed no difference compared with controls (Fig 2B). Moreover, in skin fibroblasts from patients, the level of both functional α-dystroglycan glycosylation, examined by Western blot and flow cytometry (Stevens et al, 2013b), and its ability to bind laminin (Fig 2B and Appendix Fig S5B) were normal, unlike known secondary dystroglycanopathies, which usually result in decreased functional α-dystroglycan glycosylation in both muscle and the skin fibroblasts (Willer et al, 2012; Carss et al, 2013; Stevens et al, 2013a). Finally, although basement membrane defects are commonly observed in dystroglycanopathies (Yamamoto et al, 1997; Goddeeris et al, 2013), transmission electron microscopy showed normal muscle ultrastructure in patients, with no alterations in basement membrane compaction (Fig 2C and Appendix Fig S5C).

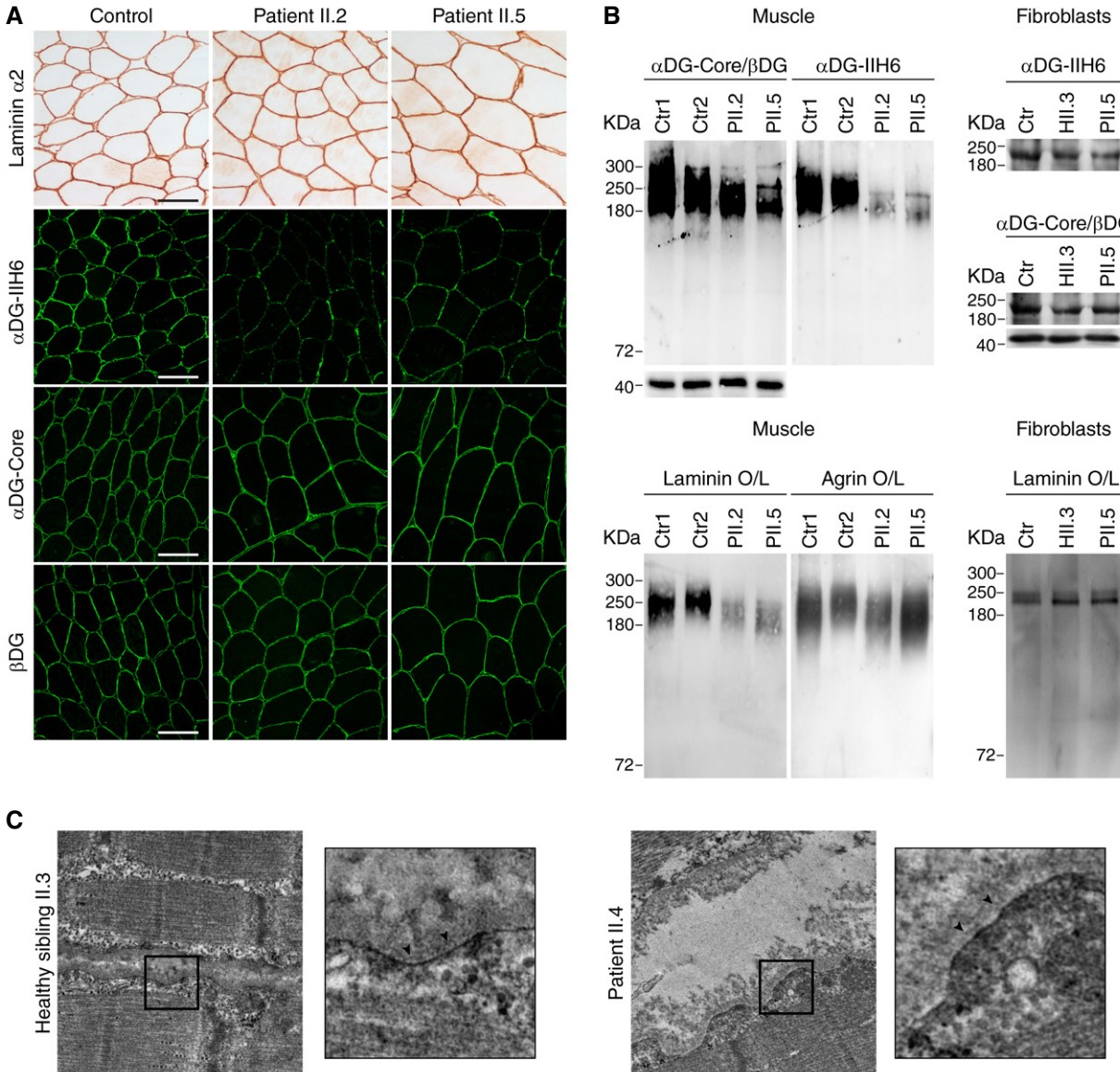

**Figure 2.  A muscle-specific decrease in α-dystroglycan glycosylation and laminin binding upon *POGLUT1* mutation.**

A   Muscle sections show variable labeling using an antibody against glycosylated α-dystroglycan (αDG-IIH6), whereas labeling using antibodies against α-dystroglycan core protein (αDG-Core), β-dystroglycan (βDG), and laminin-α2 is similar to control (scale bar, 100 μm).

B   Western blots and ligand overlay (O/L) of wheat germ agglutinin-enriched muscle and fibroblasts lysates from PII.2, PII.5, the healthy sibling (HII.3), and healthy controls (Ctr, Ctr1, and Ctr2) (muscle: 250 μg protein/lane, fibroblasts: 800 μg/lane). In muscle, expression of αDG-IIH6 is reduced, but the αDG-Core shows similar expression as controls, with a slight reduction in molecular weight; laminin overlay assay detected some binding activity but was diminished compared with controls, whereas agrin-binding activity showed no difference with controls. In fibroblasts, both αDG-IIH6 and αDG-Core expression and laminin-binding activity were normal, suggesting that unlike other muscular dystrophies, α-dystroglycan glycosylation defect is muscle specific in *POGLUT1* patients.

C   No ultrastructural alterations are observed in muscle by electron microscopy (6–7 fields were captured from PII.1, PII.3, and PII.5). Note normal basement membrane compaction (arrowheads). Original magnification: ×26,700.

Source data are available online for this figure.

These observations suggest that although a partial decrease in α-dystroglycan functional glycosylation is detected in muscle from our patients, decreased α-dystroglycan glycosylation is unlikely to be caused by the same pathomechanism seen in secondary dystroglycanopathies and that a mutation in a gene other than the enzymes directly involved in α-dystroglycan glycosylation might underlie this disease.

## D233E mutation in *POGLUT1*

Homozygosity mapping in DNA samples through genomewide genotyping arrays containing ~ one million single nucleotide polymorphisms (SNPs) revealed a unique (> 500 kb) region of homozygosity identical by descent in the four affected siblings and absent in their

healthy sister (Fig 3A). This 14.6 megabase region on chromosome 3q13.13-q21.2, defined by SNP markers rs13081374 and rs2333038, included 112 protein-coding genes (Fig 3B and C). No structural genomic variations were shared by affected family members, thus excluding gross gene dosage change as a cause of disease. We next performed whole exome sequencing of patients II.4, II.5, and their healthy sister II.3 and found a c.699T > G transversion within *POGLUT1* (Fig 3D). The polymorphism produces an aspartic-to-glutamic substitution at amino acid residue 233 (p.D233E), which passed all possible filtering, segregated with the disease, and consequently was suspected to be the causal genetic alteration.

Significantly, *POGLUT1* is located within the 3q13.13-q21.2 region of homozygosity in the affected siblings (Fig 3C). The mutation was not found in the Exome Aggregation Consortium (ExAC) database (http://exac.broadinstitute.org) [accessed in February 2015], thus discarding this genetic variant in 60,698 individuals (36,673 Europeans). In addition, the D233E mutation was further excluded in 919 DNA samples from 57 populations worldwide (Appendix Table S3).

The mutated aspartic acid residue is located in the capsule-associated protein (CAP)10 domain, 14 amino acids apart from the putative ERD catalytic motif, and is highly conserved across vertebrate species (Fig 3E).

### Expression of POGLUT1$^{D233E}$ in patients

We assessed the effect of the D233E mutation on the expression of POGLUT1 in samples from patients. In skeletal muscle, Western blot showed normal protein expression and qRT–PCR assays showed no difference in the *POGLUT1* mRNA level compared with controls. In skin biopsy-derived fibroblasts, immunostaining revealed a colocalization of POGLUT1$^{D233E}$ with the endoplasmic reticulum, displaying the same pattern as the wild-type POGLUT1 in human control fibroblasts (Appendix Fig S6). These data indicate that the D233E mutation does not affect the expression level or subcellular localization of POGLUT1.

### Biochemical assays of D233E mutation in *POGLUT1*

We next analyzed the effect of the D233E mutation on the *O*-glucosyltransferase activity of POGLUT1 by biochemical assays. Wild-type and D233E mutant forms of human-POGLUT1 were transiently expressed in HEK293T cells and purified from the culture media (Appendix Fig S7A). We examined the enzymatic activity of purified proteins toward a known substrate: bacterially expressed epidermal growth factor-like (EGF) repeat from human coagulation factor IX (hFIX). POGLUT1$^{D233E}$ mutant showed significantly lower activity than wild type (Fig 4A). *O*-glucosyltransferase activity was dependent on the concentrations of EGF repeat and UDP-glucose (acceptor and donor substrate, respectively). POGLUT1$^{D233E}$ also showed lower activity than wild type toward five different single EGF repeats from mouse Notch1 (Fig 4B), suggesting that the mutation affects *O*-glucosylation of all EGF repeats containing the *O*-glucose consensus sequence (CXSX(P/A)C) (Rana *et al*, 2011). Prior work has shown that POGLUT1 also has protein *O*-xylosyltransferase activity toward certain EGF repeats with a diserine motif within the *O*-glucose consensus sequence (e.g. EGF16 of mouse Notch2) (Takeuchi *et al*, 2011). POGLUT1$^{D233E}$ mutant also showed lower

*O*-xylosyltransferase activity than wild type (Appendix Fig S7B). To examine whether POGLUT1$^{D233E}$ retains any enzymatic activity, we performed overnight incubation and then analyzed the products by reverse-phase HPLC and mass spectrometry. Both wild type and the D233E mutant added a single glucose or xylose to the hFIX-EGF repeat (Fig 4C) or EGF16 from mouse Notch2 (Appendix Fig S7C), indicating that POGLUT1$^{D233E}$ has residual enzymatic activity.

### Analysis of Notch signaling pathway and satellite cells in D233E patients

Mutations in the *Drosophila* gene *rumi*, which encodes the fly-POGLUT1, result in a temperature-sensitive loss of Notch signaling (Acar *et al*, 2008). Moreover, *Poglut1* knockdown in mouse myoblast C2C12 cells results in a significant decrease in *O*-glucosyltransferase activity and impaired Notch signaling (Fernandez-Valdivia *et al*, 2011). Accordingly, we examined the expression of activated Notch and the levels of Notch downstream targets in patient muscles. Decreases in Notch1-intracellular domain expression and in *HES1* mRNA levels in skeletal muscle were detected compared with controls, which support that D233E affects Notch activity (Fig 5A and B). Notch signaling is critical in normal postnatal myogenesis by regulating SC maintenance (Bjornson *et al*, 2012), at least in part by directly inducing the expression of *Pax7*, which encodes a transcription factor critical for the maintenance of adult SC (Olguin & Olwin, 2004). Indeed, treating proliferating C2C12 cells with the γ-secretase inhibitor DAPT, which blocks Notch signaling, significantly reduced the mRNA levels of *Pax7* (Appendix Fig S8). Likewise, we found a decrease in *PAX7* mRNA and fewer PAX7$^{+}$ cells in the muscle of patients compared with controls (Fig 5C and D). Together, these observations suggest that the D233E mutation affects the self-renewal of SC by reducing Notch signaling, likely due to defective *O*-glucosylation.

We analyzed the same parameters in patients suffering from other muscular dystrophies, and also in patients suffering from a secondary dystroglycanopathy, whose muscle biopsies showed mild-to-moderate dystrophic changes, similar to our patients. Both groups show significantly higher levels of activated Notch and higher numbers of PAX$^{+}$ cells compared with D233E patients (Appendix Fig S9A–H). These data support the notion that the observed decrease in Notch1 signaling and PAX7$^{+}$ cells in D233E patients is not secondary to α-dystroglycan hypoglycosylation.

### *In vivo* functional study of POGLUT1$^{D233E}$

To investigate the *in vivo* effects of the D233E mutation on POGLUT1 function in muscle progenitors, we performed cross-species rescue experiments in the context of adult myogenesis in *Drosophila*, a process regulated by Notch signaling (Gildor *et al*, 2012). In these experiments, we used the *rumi*$^{79}$ allele, which harbors a missense mutation that abolishes fly-POGLUT1 enzymatic activity but does not affect its expression (Acar *et al*, 2008). When raised at 18°C, *rumi*$^{79/79}$ animals exhibited normal indirect flight muscle development and a large number of myoblasts expressing the transcription factor Twist, which is a downstream target of Notch activation (Fig 6A–B″). However, when *rumi*$^{79/79}$ animals were raised at 25 and 30°C during the early pupal stage, when these muscles form, muscle development was dramatically impaired and

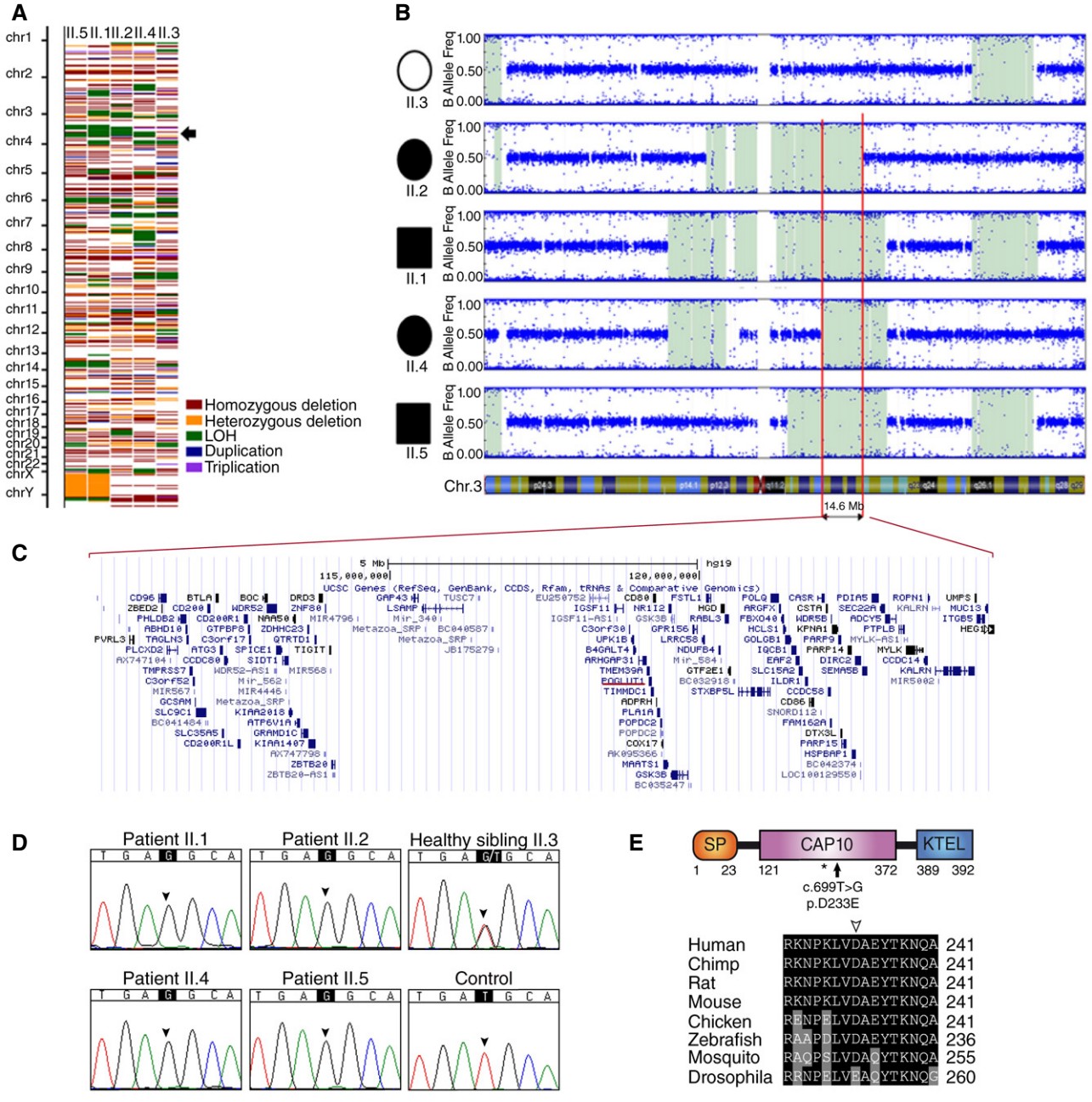

**Figure 3.  *POGLUT1* D233E mutation in patients.**

A   Genomewide mapping through high-density genotyping. The structural genomic variation and extended regions of homozygosity (loss of heterozygosity [LOH]) over the entire genome is shown for individuals II.1 to II.5. The homozygosity segment shared by the four affected siblings in chromosome 3 is indicated with an arrow.

B   This region is represented in more detail, where extended regions of homozygous genotypes in contiguous biallelic polymorphisms are highlighted with a green background.

C   The 14.6 megabase region of homozygosity that is shared by all affected siblings is highlighted by vertical red lines in (B), and genes within this genomic region are shown. Note that *POGLUT1* gene is underlined.

D   Chromatograms reveal a homozygous T-to-G substitution (black boxes and arrowheads) in the four affected family members, while a heterozygous G/T in the healthy sibling.

E   Schematic structure of the human POGLUT1 protein, which contains a signal peptide (SP), a CAP10 domain, and a Lys-Asp-Glu-Leu (KDEL)-like endoplasmic reticulum retention signal. D233E is located in the CAP10 domain (arrow), close to the predicted ERD catalytic motif (asterisk). Alignment of amino acids flanking the mutated aspartic acid from POGLUT1 orthologs indicates evolutionary conservation of D233 in vertebrates (open arrowhead). Black boxes denote similarity of amino acid residues.

the number of Twist⁺ myoblasts was severely decreased (Fig 6C–D″). The severe temperature-sensitive defects observed in indirect flight muscle development in *rumi79/79* animals indicate that adult

myoblast development in flies depends on the enzymatic activity of fly-POGLUT1. Overexpression of wild-type human-POGLUT1-FLAG in muscle progenitors rescued the muscle phenotypes of *rumi79/79*

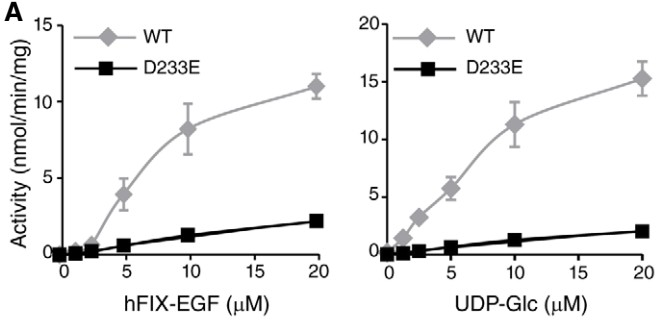

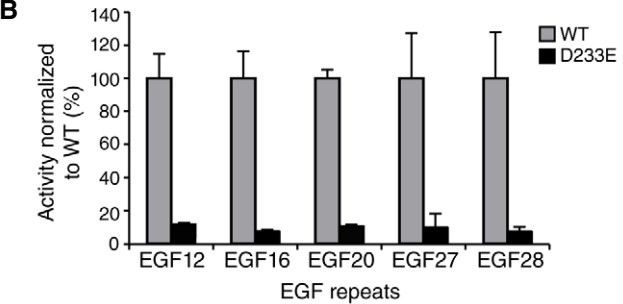

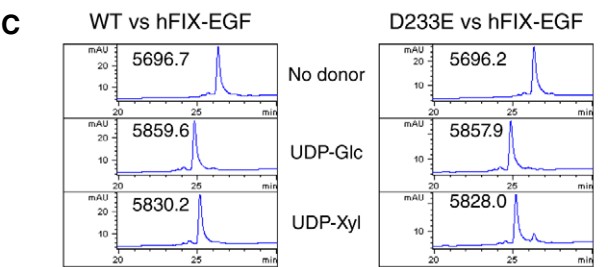

**Figure 4.  POGLUT1<sup>D233E</sup> has lower enzymatic activity than wild type.**

A   Protein *O*-glucosyltransferase activity of wild-type (WT) or D233E mutant POGLUT1 protein toward human factor IX EGF repeat (hFIX-EGF). Wild type shows higher *O*-glucosyltransferase activity than D233E, and this activity is dependent on the concentration of the acceptor substrate hFIX-EGF repeat (left) and on the concentration of donor substrate UDP-glucose (UDP-Glc) (right). Values indicate mean ± SEM from three independent assays.

B   *O*-Glucosyltransferase activity toward five different single EGF repeats from mouse Notch1. Wild-type POGLUT1 shows higher activity toward all EGF repeats tested than POGLUT1<sup>D233E</sup>. The *y*-axis shows the normalized activity relative to wild-type POGLUT1. EGF repeats were at 10 μM. Values indicate mean ± SEM from three independent assays.

C   Elution profiles of the POGLUT1 reaction products on reverse-phase HPLC. hFIX-EGF repeat was incubated with wild-type or D233E mutant POGLUT1 and donor substrate, UDP-Glc or UDP-xylose (UDP-Xyl), at 37°C overnight. Values on top of the peaks indicate the measured masses. Addition of Glc (162 Da) or xylose (132 Da) to hFIX-EGF (5696.2 Da) by wild-type POGLUT1 caused a shift to an earlier retention time (left). Similarly, the products exhibited a similar shift after incubation with POGLUT1<sup>D233E</sup> (right). These results indicate that POGLUT1<sup>D233E</sup> can add a single glucose or xylose to hFIX-EGF repeats and thus has residual enzymatic activity.

animals (Fig 6E). However, the overexpression of human-POGLUT1<sup>D233E</sup>-FLAG only showed a weak and variable rescue of the muscle morphology and myoblast numbers in *rumi*<sup>79/79</sup> animals (Fig 6F–H), indicating that this mutation significantly decreases the ability of POGLUT1 to promote muscle development and Notch signaling in myoblasts. It is interesting to note that fly-POGLUT1

harbors a glutamic acid (E) at position 252, which is equivalent to D233 in human-POGLUT1 (Fig 3E). Given the robust protein *O*-glucosyltransferase activity of the fly-POGLUT1 and because 35% of the amino acids in the CAP10 enzymatic domains of human and fly POGLUT1 are divergent (Acar *et al*, 2008), other amino acid differences between these two proteins are likely to be the reason why the fly protein is fully functional despite harboring the equivalent of the D233E mutation.

### *Ex vivo* analysis of D233E primary myoblasts

Previous studies have shown that the Notch signaling pathway not only plays an important role in maintaining satellite cell quiescence, but also enhances myoblast proliferation and inhibits myogenic differentiation (Conboy & Rando, 2002; Bjornson *et al*, 2012; Mourikis & Tajbakhsh, 2014).

In order to directly test our hypothesis that the decrease in the POGLUT1 enzymatic activity caused by the D233E mutation disrupts muscle cell function, we explored myogenesis in D233E primary myoblasts from patient muscle biopsies compared with healthy and disease controls (Appendix Fig S10). We quantified the percentage of self-renewing, proliferating, and differentiating cells based on PAX7 and MyoD expression. Quantitative analysis in myoblasts cultured in a growth medium revealed that the percentage of PAX7<sup>+</sup>MyoD<sup>+</sup> (proliferating) cells in D233E cells was significantly lower than those in healthy and disease controls (Fig 7A–C). The percentage of PAX7<sup>+</sup> cells and PAX7<sup>+</sup>MyoD<sup>−</sup> (self-renewing) cells was significantly decreased in D233E samples (Fig 7D and E), whereas the percentage of PAX7<sup>−</sup>MyoD<sup>+</sup> (differentiating) cells was increased (Fig 7F). When myoblasts reached confluence, they were cultured in a differentiation medium. We observed more efficient myoblast fusion into the multinucleated myotubes and higher levels of myogenin expression in D233E cultures compared with controls (Fig 7G–I). The slow proliferation and facilitated differentiation of mutant myogenic cells are in agreement with decreased Notch activity in the muscle of D223E patients.

Due to the scarcity of cells obtained from fresh patient's muscles in culture (likely because of the severe defect in proliferation of D233E myoblasts), we performed immortalization of the primary myoblasts (Mamchaoui *et al*, 2011), and conversion of skin fibroblasts from patients to myogenic cells by transduction of inducible MyoD (Choi *et al*, 1990). Thus, we generated two different continuous myogenic cell lines with the D233E mutation, which showed similar phenotypic features, although PAX7 expression started after 1–3 days in culture in the myogenic cell line (Appendix Figs S11 and S12). To examine whether increasing Notch pathway activity can rescue the alterations of the myogenic process observed in D233E immortalized myoblasts during differentiation in culture, we used lentivirus to overexpress NICD1 in these cells and observed a solid reversion of the proliferation defects (Fig 8A and B, Appendix Fig S13A and B). However, we did not observe rescue of the amount of PAX7<sup>+</sup> cells (Fig 8C). It has been described that the effect of Notch on the expression of PAX7 in myogenic cell lines such as C2C12 is not as potent as in primary myoblasts (Sun *et al*, 2008). This probably hindered the modification of PAX7 expression during the rescue experiment. In addition, NICD1 overexpression robustly rescued the facilitated differentiation shown by D233E myoblasts in culture (Fig 8D, Appendix Figs S13C and S14), and this

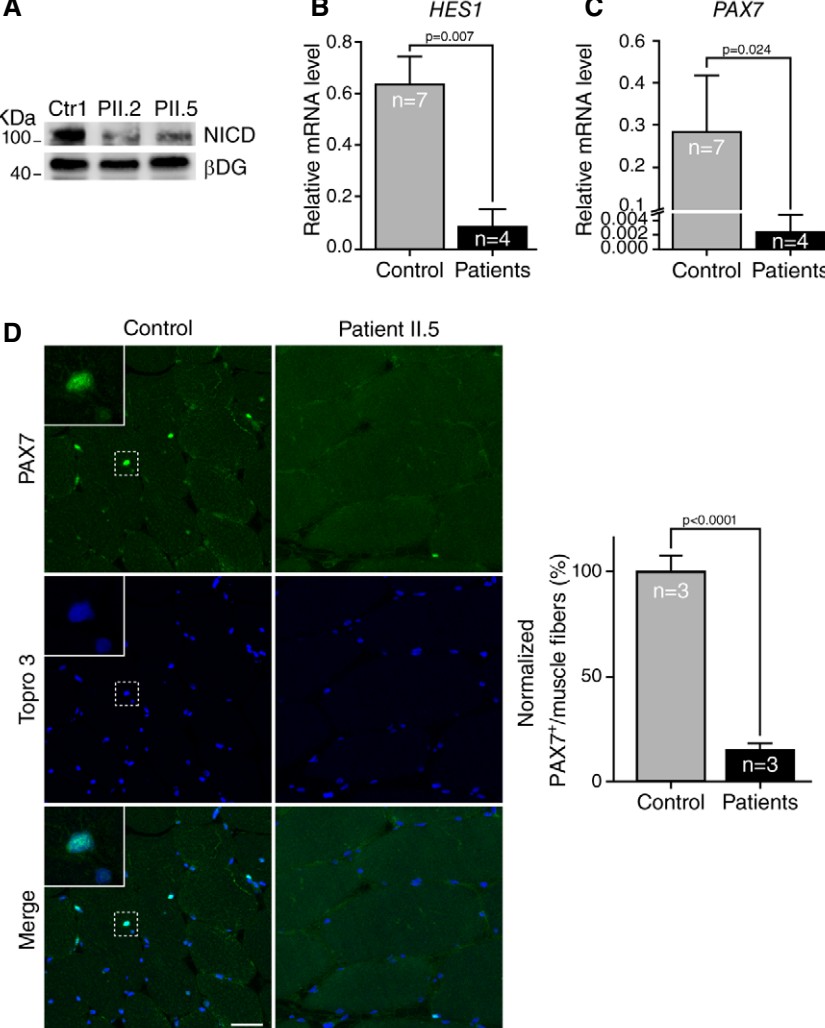

**Figure 5.  Muscles from D233E patients show decreased Notch signaling and pool of satellite cells.**

A    Western blot of muscle homogenates shows reduced expression of Notch1 intracellular domain (NICD) in patients.

B, C   qRT–PCR on RNA extracted from skeletal muscle shows that expression of *HES1* (B) and *PAX7* (C) was significantly lower in D233E patients compared with healthy controls. Mean ± SEM; Student's *t*-test (B) and Mann–Whitney *U*-test (C).

D    PAX7+ cells in skeletal muscle sections, demonstrating that satellite cells are less abundant in D233E patients than in controls (*n* = 3 muscles with 10–15 fields analyzed per muscle). Mean ± SEM; Mann–Whitney *U*-test; scale bar, 50 μm.

Source data are available online for this figure.

rescue was demonstrated by the fusion index (Fig 8E) and expression of myogenin (Fig 8F). These results support the pathogenic role of decreased Notch signaling in this novel muscular dystrophy.

**Role of Notch on α-dystroglycan glycosylation**

As shown in Fig 2, our patients exhibit a muscle-specific reduction in α-dystroglycan glycosylation. The only known protein domain to which POGLUT1 can add carbohydrates is a properly folded EGF repeat containing the CX<u>S</u>X(P/A)C consensus sequence (Takeuchi *et al*, 2012). Since α-dystroglycan contains no EGF repeats nor CXSX (P/A)C sites, it is unlikely to be a direct target of glycosylation by POGLUT1. A recent study has shown that in transgenic mice and

C2C12 myoblasts, α-dystroglycan glycosylation is a progressive process that initiates rapidly after myoblasts are induced to differentiate, and exhibits a regular and gradual transition to a larger glyco-forms to reach the normal level after 5 days of differentiation (Goddeeris *et al*, 2013). In order to evaluate this progressive pattern of α-dystroglycan glycosylation, primary immortalized myoblasts were differentiated. After 5 days with differentiation medium, D233E myoblasts showed a lower level and irregular progression profile of α-dystroglycan glysosylation compared with controls, which reproduced the same pattern previously demonstrated in C2C12 cells (Goddeeris *et al*, 2013) (Fig 9A). These observations are compatible with abnormal differentiation dynamics in patient myoblasts. Given the well-established role of Notch signaling in

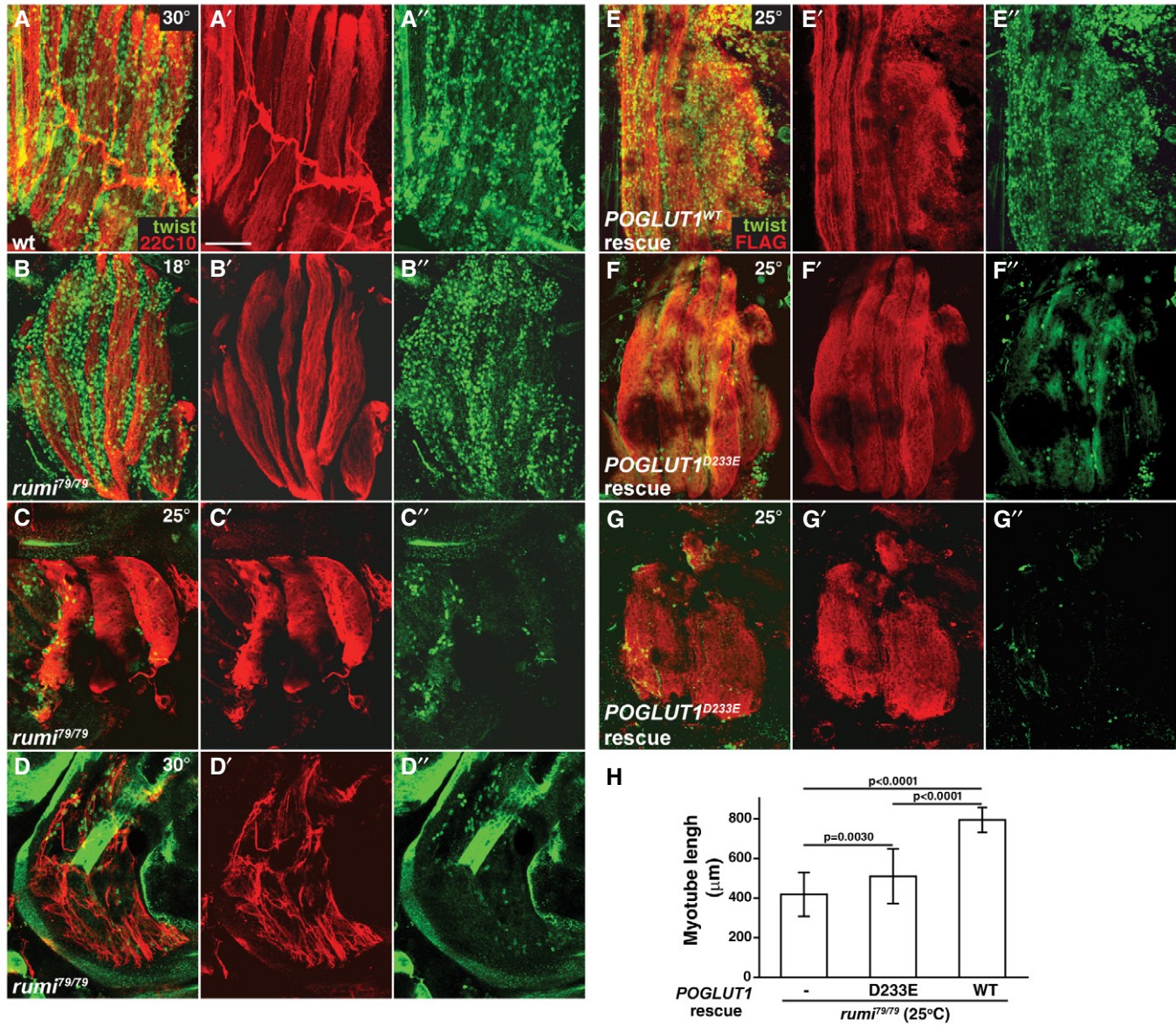

**Figure 6. Loss of Rumi results in temperature-sensitive *Drosophila* muscle defects and is only weakly rescued by POGLUT1^D233E expression.**

A–A″   Staining of *Drosophila* indirect flight muscles at 25% pupal development with 22C10 antibody (myotube marker) and anti-Twist antibody (myoblast marker) indicates that in wild-type animals, indirect flight muscles are formed from six myotubes, and a large number of Twist[+] myoblasts are present.

B–B″   *rumi*^79/79 mutants raised at 18°C show no obvious defects in myoblast number or myotube morphology.

C–D″   *rumi*^79/79 mutants exhibit decreased number of myoblasts at 25–30°C and either an incomplete set of short myotubes when raised at 25°C or aberrant morphology of myotubes when raised at 30°C.

E–E″   *rumi*^79/79 animals overexpressing FLAG-tagged POGLUT1^WT in the muscle compartment using *Mef2-GAL4* and raised at 25°C during the pupal stage show a rescue of the *rumi*^79/79 muscle phenotype.

F–G″   Overexpression of FLAG-tagged POGLUT1^D233E only partially rescues the *rumi*^79/79 muscle phenotype, with some variation in the myotube length and the number of restored Twist[+] cells (compare G and F). Note that the number of Twist[+] cells restored by POGLUT1^D233E is much smaller than that restored by POGLUT1^WT.

H   Quantification of myotube lengths in *rumi*^79/79 mutants with or without POGLUT1 expression indicates a weak rescue of the phenotypes by POGLUT1^D233E compared with POGLUT1^WT. The differences in myotube lengths are statistically significant. For *rumi*^79/79 rescue (left bar), 5 animals were used and a total of 23 myotubes were measured; for D233E rescue, 11 animals and 54 myotubes; for WT rescue, 5 animals and 22 myotubes. Mean ± SD is shown; one-way ANOVA with Bonferroni's multiple comparisons test.

Data information: Scale bar in (A′) is 50 μm and applies to (A–G″).
Source data are available online for this figure.

muscle differentiation, we hypothesized that decreased Notch signaling in D233E patients' myoblasts might contribute to altered α-dystroglycan glycosylation. We first attempted to test this hypothesis by asking whether NICD overexpression can restore α-dystroglycan glycosylation in patient myoblasts. However, given the strong inhibition of differentiation upon NICD overexpression,

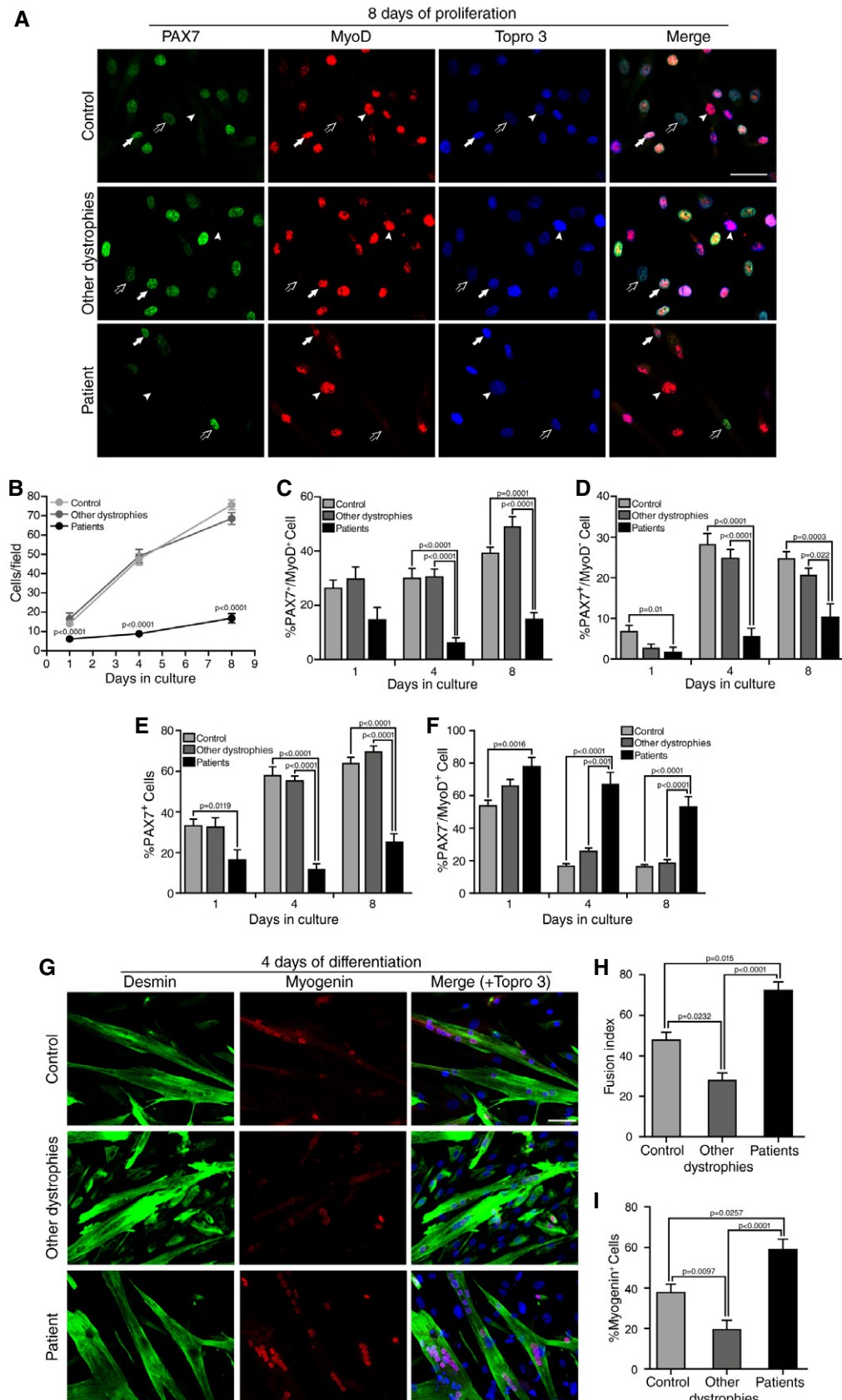

Figure 7.

**Figure 7.  D233E primary myoblasts exhibit decreased proliferation and increased differentiation.**

A–F   Proliferation assays examined proliferating (PAX7+ MyoD+, arrows), self-renewing (PAX7+ MyoD−, open arrows), and differentiating (PAX7− MyoD+, arrowheads) cells from D233E patients (n = 2), healthy controls (n = 3), and disease controls (n = 3) at 1, 4, and 8 days growing in proliferation medium. Panel (A) shows representative images from day 8. The number of patient myoblasts is lower than that of controls at all three time points, indicating a slower proliferation rate in patient myoblasts (B), and accordingly, the percentage of proliferating cells (C) is smaller in the patient. (D) The percentage of self-renewing cells is small in patient's culture, reflecting a poor capacity for maintaining the pool of quiescent SCs. In addition, the percentage of PAX7+ cells (E) is smaller in patient's culture, similar to what was found in adult D233E muscle (shown in Fig 5D). (F) The percentage of differentiating cells is higher in the patient. We examined proliferation capturing 7–12 randomly fields per day and condition (mean ± SEM is shown; Kruskal–Wallis with Dunn's multiple comparisons test; scale bar, 50 μm).

G–I   When myoblasts started to be confluent, the proliferation medium was replaced with differentiation medium, and the myoblasts started to fuse yielding myotubes, which were analyzed after 4 days of differentiation. (H) The fusion index, which measures the percentage of nuclei in myotubes with respect to the total number of nuclei in myogenic cells, was higher in D233E cultures than in controls. (I) Besides, myogenin showed increased expression in D233E myoblast cultures compared with controls. The data support that differentiation process is facilitated in patient's muscle. Data were collected from 3 to 10 randomly chosen areas per condition (mean ± SEM is shown; Kruskal–Wallis with Dunn's multiple comparisons test (H) and one-way ANOVA with Bonferroni's multiple comparisons test (I); scale bar, 50 μm).

Source data are available online for this figure.

we were unable to do this, as in both cases (rescued and not rescued patient's myoblasts) α-dystroglycan glycosylation was immature (Fig 9B). Next, we examined whether inhibition of Notch signaling leads to altered pattern of α-dystroglycan glycosylation during differentiation. To this end, we differentiated C2C12 cells with medium containing DAPT or LY3039478 to decrease Notch signaling. The cells showed a reduced expression of glycosylated α-dystroglycan at all differentiation days (Fig 9C–F). These results support the notion that α-dystroglycan hypoglycosylation is the consequence of Notch inactivation, produced by the reduced activity of POGLUT1 in the case of our patients.

## Discussion

Transgenic mice in which the pan-Notch inhibitor dnMAML1 is specifically expressed in muscle SC were recently shown to display reduced PAX7 expression in SC cells and muscular dystrophic features even without experimentally induced injury (Lin *et al*, 2013). This study highlighted the role of Notch signaling and SCs in repairing the muscle damage caused by regular activity in a mammal, suggesting that a primary defect in SC maintenance caused by decreased Notch signaling can in principle result in muscular dystrophy. However, direct evidence from human patients was lacking to support this notion. Here, we describe a family in which a recessive D233E mutation severely impairs the enzymatic activity of POGLUT1, leading to a marked reduction in Notch activation in adult muscle, a decreased pool of PAX7+ SCs, and a partial defect of α-dystroglycan functional glycosylation. The findings in this family sharply contrast with those in other forms of

age-matched limb-girdle muscular dystrophy, including secondary dystroglycanopathies, which show normal levels of *HES1* and *PAX7* transcripts in adult muscles (Appendix Fig S9).

Although patients' muscles display reduced glycosylation of α-dystroglycan, several findings are inconsistent with a secondary dystroglycanopathy. First, the patients do not have elevated creatine kinase and muscle biopsies do not show many degenerative fibers compared with that observed in secondary dystroglycanopathies (Cirak *et al*, 2013). Second, the D233E muscles exhibit reduced α-dystroglycan laminin-binding activity but show normal agrin-binding and compact basement membrane ultrastructure. Although it is known that LARGE-dependent glycosylation of α-dystroglycan is required for the binding of both laminin and agrin, the precise glycan structures recognized by these ligands have not been determined. Agrin, but not laminin, showed normal binding to α-dystroglycan in patients' muscle, suggesting that these ligands may recognize slightly different glycan structures on α-dystroglycan. Third, α-dystroglycan expression and laminin binding are normal in D233E skin fibroblasts. In contrast, other forms of dystroglycanopathy usually exhibit markedly diminished or no binding for both laminin and agrin in muscle, together with defective muscular basement membrane compaction (Michele *et al*, 2002; Hara *et al*, 2011; Willer *et al*, 2012; Goddeeris *et al*, 2013). Moreover, the published results show that secondary dystroglycanopathies caused by mutations in known genes result in reduced functional α-dystroglycan glycosylation in patient fibroblasts, and in the majority of patients, the level of α-dystroglycan glycosylation is comparable in fibroblasts and skeletal muscle from the same patient (Carss *et al*, 2013). These observations further support the notion that, although decreased laminin binding in muscle is likely to contribute to pathology in our

**Figure 8.  Rescue experiments on immortalized primary myoblasts by lentiviral infection.**

A–C   We overexpressed NICD in immortalized primary myoblasts to rescue the features displayed by immortalized D233E myoblasts. (B) We analyzed the proliferation at 1, 3, and 5 days, and proliferation rate in patient-LV-NICD-GFP reached the level observed in control-LV-GFP, which were higher than in patient-LV-GFP. (C) However, the percentage of PAX7+ cells in patient-LV-NICD-GFP showed the same reduced value as in patient-LV-GFP, which were significantly lower than in control-LV-GFP. The majority of control myoblasts were proliferating cells (PAX7+ MyoD+, arrows), with no quiescent cells and few differentiating cells (PAX7− MyoD+, arrowheads) detected. Data were collected from n = 2 independent assays (5–8 images per condition). Mean ± SEM is shown; one-way ANOVA with Tukey multiple comparisons test (B) and Kruskal–Wallis with Dunn's multiple comparisons test (C); scale bar, 50 μm.

D–F   As previously described in patient's primary myoblasts, patient-LV-GFP myoblasts showed a striking facilitated differentiation compared with control-LV-GFP myoblasts. This phenotype fully disappeared in patient-LV-GFP-NICD myoblasts, as quantification of the fusion index (E), and myogenin expression (F) demonstrated. The data support a pathogenic role for Notch inhibition on the altered differentiation in our patients. Data were collected from n = 3 independent assays (13–17 images per condition). Mean ± SEM is shown; Kruskal–Wallis with Dunn's multiple comparisons test; scale bar, 50 μm.

Source data are available online for this figure.

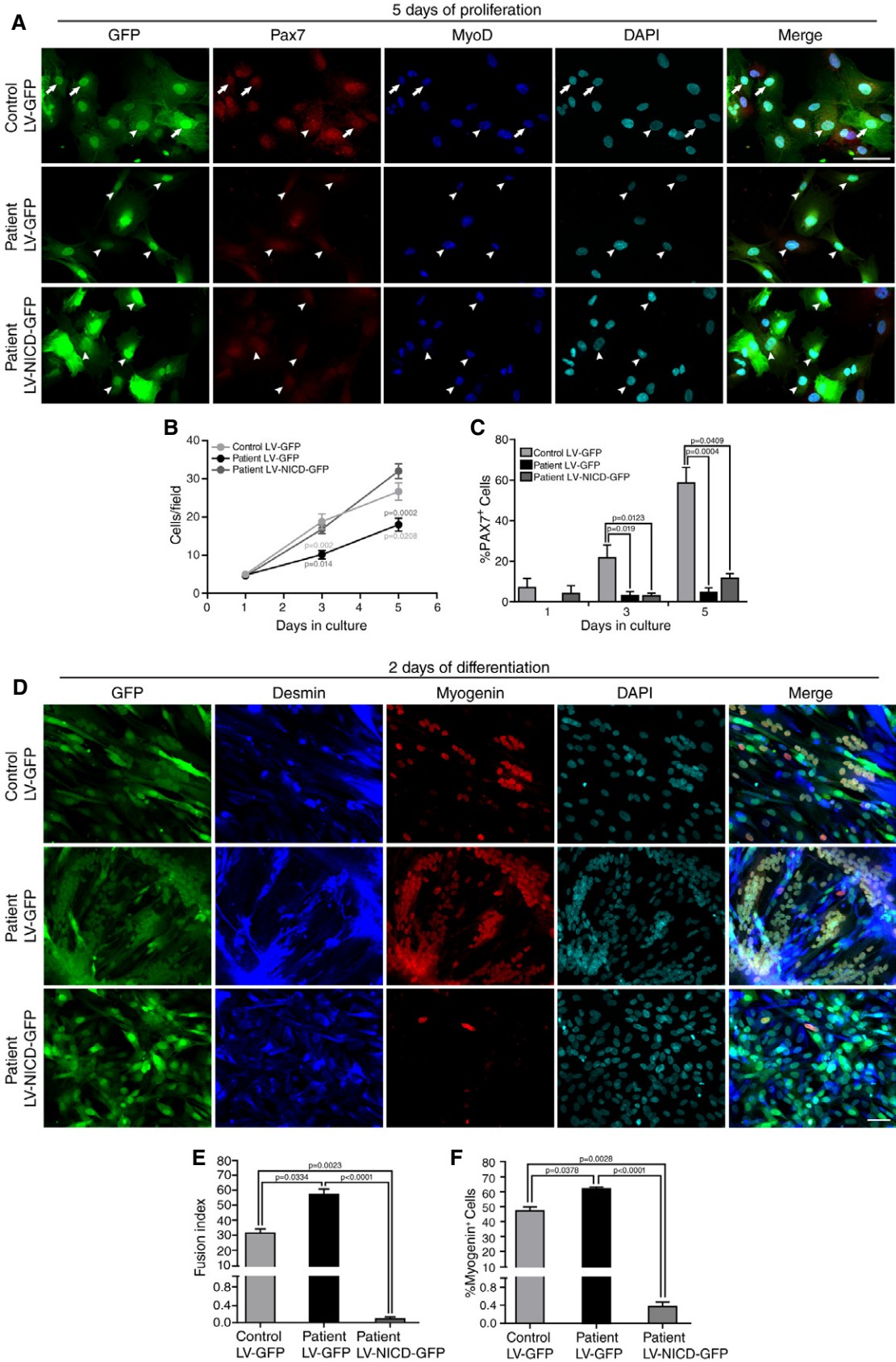

Figure 8.

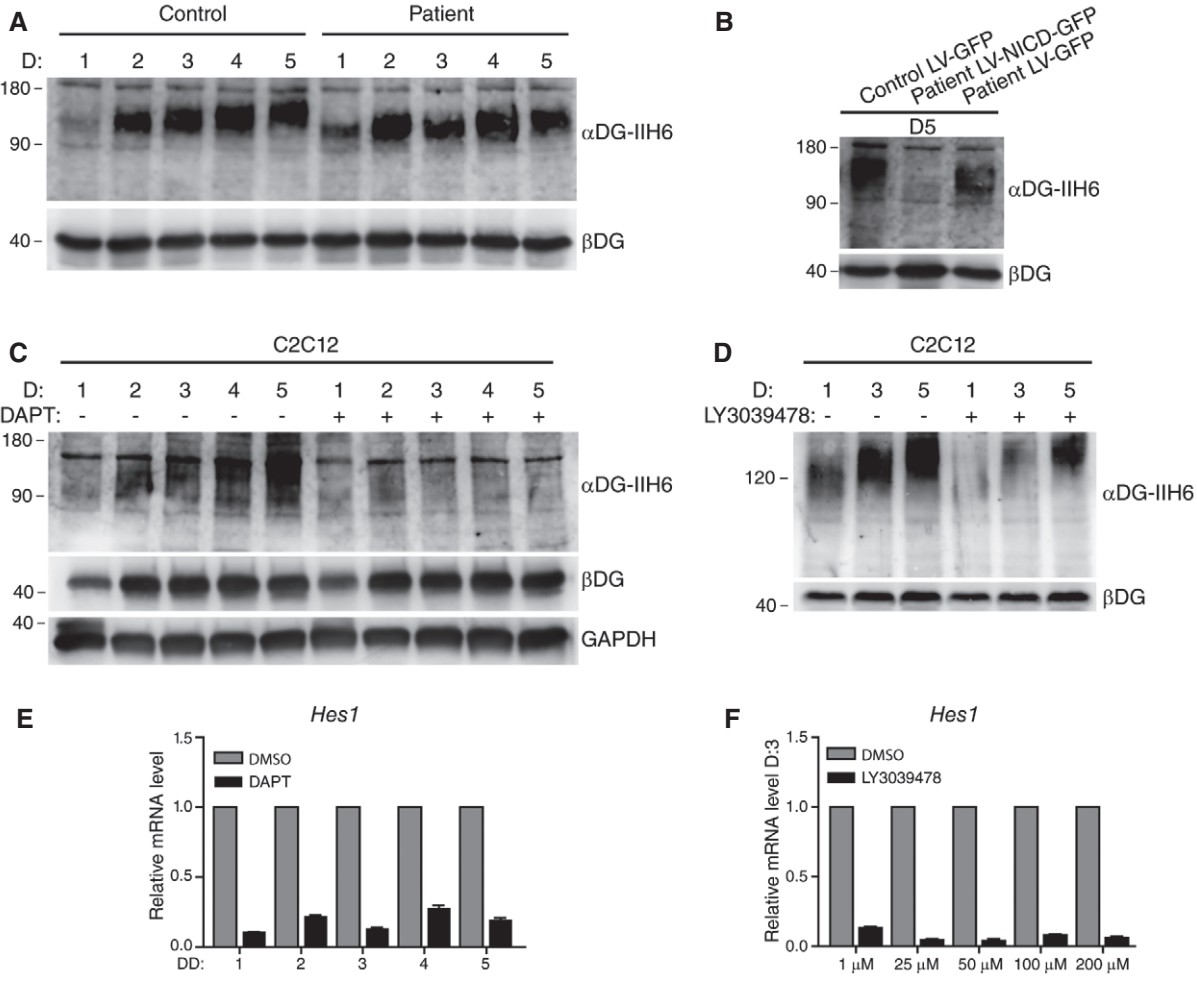

**Figure 9.  Effects of decreased Notch signaling on α-dystroglycan glycosylation during differentiation.**

A    Immortalized primary myoblasts from a healthy control showed a progressive pattern of α-dystroglycan glycosylation during 5 days of differentiation (D0–D5), as has been previously described by Goddeeris *et al* (2013). Immortalized primary myoblasts from patients displayed an irregular pattern of glycosylation, with a final level of glycosylated α-dystroglycan that was lower than control.

B    α-dystroglycan glycosylation of infected immortalized myoblasts from patient-LV-NICD-GFP at differentiation day 5 (D5) showed a striking reduction, in line with the strong inhibition of myotubes formation and differentiation induced by NICD overexpression. α-dystroglycan in myoblasts from patient-LV-GFP showed a reduction in glycosylation compared with control-LV-GFP at D5, similar to the results obtained in the adult muscle from patients (Fig 2).

C, D  C2C12 myogenic cells showed a reduced level of glycosylated α-dystroglycan when the Notch signaling inhibitor DAPT or LY3039478 was added to the differentiation medium. This indicates that reducing Notch signaling can affect α-dystroglycan glycosylation in wild-type C2C12 myoblasts cells, which do not harbor any known mutations in α-dystroglycan glycosyltransferases or Poglut1.

E, F  qRT–PCR experiments show that upon DAPT or LY3039478 treatment, a remarkable decrease in *Hes1* mRNA is induced in C2C12 cells during differentiation. Three replicates per condition were analyzed; mean ± SEM is shown.

Source data are available online for this figure.

patients, α-dystroglycan hypoglycosylation might not be the disease mechanism caused by the D233E mutation, or at least the primary mechanism.

Strikingly, our data suggest that the D233E mutation causes defects in SC proliferation and differentiation. The amount number of SCs in our patients is significantly reduced even in very early stages when the muscle is barely affected, compared with healthy and disease controls. Alterations in SC proliferation have been described in the Large[myd] mouse dystroglycanopathy model and are thought to contribute to myodegeneration in these animals (Ross *et al*, 2012). In this model, SCs are significantly increased at early

stages and have reduced proliferation when cultured on freshly isolated single fibers; however, proliferation is restored in culture when SCs are removed from the fibers, indicating that rather than intrinsic defects in myoblasts, compositional changes in the extracellular matrix due to hypoglycosylated α-dystroglycan are responsible for the proliferative alteration. In contrast, myoblasts from our patients displayed reduced proliferation and self-renewal capacity in culture when removed from their fiber niche, supporting an intrinsic defect in D233E myoblasts independent of α-dystroglycan hypoglycosylation. Moreover, it has been recently demonstrated in dystroglycanopathy mice models that functional glycosylation of

α-dystroglycan does not play a significant role in myogenic cell proliferation and differentiation (Awano *et al*, 2015). Our results indicate that the D233E mutation results in deficient Notch signaling in patient muscle and an impairment in the regulation of myoblast differentiation and that expression of NICD rescues this phenotype. Our results strongly suggest that impaired myogenesis mediated by decreased Notch signaling is a causal mechanism of muscular dystrophy in our patients (Fig 8). Although we expected that the reduction in PAX7[+] cells in our patients would be a direct effect of Notch inhibition, the lack of rescue of PAX7[+] cells by NICD overexpression suggests something more complicated. The effect of Notch on PAX7 expression has been reported to be weaker in myogenic cell lines than in primary myoblasts, and this could be affecting the rescue of PAX7[+] cells in our system (Sun *et al*, 2008). Nevertheless, our data consistently demonstrated a decreased number of PAX7[+] cells in D233E patients, compared with healthy and disease controls, including secondary dystroglycanopathies. This result together with a previous report showing that the blockade of Notch signaling in muscle SCs of a mouse model causes muscular dystrophy and impairs muscle regeneration (Lin *et al*, 2013) supports our hypothesis that impaired regulation of muscle SC homeostasis plays a pathogenic role in this new muscle dystrophy. Recent findings demonstrate that the overexpression of Jagged1, a major ligand of the Notch signaling pathway, ameliorates the dystrophic phenotype in the golden retriever muscular dystrophic dogs, which is an excellent model to the human dystrophynopathy, indicating that the Notch pathway can be considered a new therapeutic target in muscular dystrophies (Vieira *et al*, 2015).

POGLUT1 specifically adds *O*-glucose or *O*-xylose to properly folded EGF repeats containing the CXSX(P/A)C consensus sequence (Rana *et al*, 2011; Takeuchi *et al*, 2011, 2012) and is among the glycosyltransferases that directly add sugar molecules to an amino acid, not to other sugars. Therefore, since α-dystroglycan lacks EGF repeats and the CXSX(P/A)C consensus sequence, POGLUT1 is highly unlikely to directly add carbohydrates to α-dystroglycan core protein or to extend the carbohydrate chains added to α-dystroglycan by other glycosyltransferases. This is supported by our observation that glycosylation of α-dystroglycan from D233E skin fibroblasts is normal. These data strongly argue that POGLUT1 does not participate in the biosynthesis of the *O*-mannose glycans on α-dystroglycan. Therefore, unlike POMT1/2 and LARGE which add carbohydrates to α-dystroglycan to regulate muscle maintenance, POGLUT1 most likely glycosylates other target proteins to maintain muscle integrity. This notion is supported by our fly experiments: Loss of POGLUT1 enzymatic activity in *Drosophila* also causes a severe decrease in the number of Twist[+] myoblasts and muscle development, which can be fully rescued by wild-type human-POGLUT1 but only weakly so by POGLUT1[D233E]. However, *Drosophila* dystroglycanopathy models, including null alleles of *dystroglycan*, do not have defects in myoblast migration and myofiber formation (Nakamura *et al*, 2010), indicating that the muscle phenotypes observed in *rumi* flies are not caused by alterations in dystroglycan.

Fourteen *Drosophila* proteins and about 50 mammalian proteins have EGF repeats with a CXSX(P/A)C consensus sequence and are therefore predicted to be modified by POGLUT1 (Jafar-Nejad *et al*, 2010; Rana *et al*, 2011; Haltom *et al*, 2014). This consensus motif is highly predictive of enzymatic modification, as mass spectrometric analyses have demonstrated that most if not all EGF repeats with a

CXSX(P/A)C motif examined so far are *O*-glucosylated (Hase *et al*, 1988; Acar *et al*, 2008; Fernandez-Valdivia *et al*, 2011; Rana *et al*, 2011; Lee *et al*, 2013; Haltom *et al*, 2014; Ramkumar *et al*, 2015; Thakurdas *et al*, 2016). Agrin, which is implicated in muscle disease, is predicted to have a single POGLUT1 target site. However, we did not observe any defects in agrin's level, molecular size, or its capacity to bind α-dystroglycan in our patient muscles, strongly suggesting that impaired agrin glycosylation cannot explain the muscular dystrophy in our patients. Indeed, the addition of *O*-glucose to a target protein by POGLUT1 does not necessarily indicate functional importance. For example, even though *Drosophila* Crumbs is *O*-glucosylated, homozygosity for a knock-in allele of the *Drosophila crumbs* in which all seven Crumbs *O*-glucosylation sites are mutated does not recapitulate *crumbs* loss-of-function phenotypes during fly development and generates viable and fertile adults (Haltom *et al*, 2014). Among the confirmed POGLUT1 biochemical targets, *Drosophila* Notch, mouse Notch1, mouse CRUMBS2, and *Drosophila* Eyes shut are shown to be functionally regulated by POGLUT1-mediated glycosylation (Acar *et al*, 2008; Fernandez-Valdivia *et al*, 2011; Leonardi *et al*, 2011; Haltom *et al*, 2014; Ramkumar *et al*, 2015). Moreover, there is evidence suggesting that mouse JAG1 might also be a biologically relevant target of POGLUT1, although further experiments are required to prove this (Thakurdas *et al*, 2016). Based on the severe decrease in Notch signaling and Notch1 cleavage in our patients' muscle, the dramatic decrease in the ability of POGLUT1[D233E] to glycosylate Notch EGF repeats, and previous observations on the role of POGLUT1 and Notch receptor *O*-glucosylation in Notch signaling (Acar *et al*, 2008; Fernandez-Valdivia *et al*, 2011; Leonardi *et al*, 2011; Ma *et al*, 2011; Rana *et al*, 2011), we propose that the Notch receptors are the key target of POGLUT1 in the muscle. However, we cannot exclude that other POGLUT1 targets might also contribute to the phenotypes observed in our patients.

If POGLUT1 affects a glycosylation pathway unrelated to the *O*-mannose glycans, it raises the question of why *O*-mannose glycosylation on α-dystroglycan is impaired in D233E muscle. Our data strongly favor an indirect mechanism for α-dystroglycan hypoglycosylation, likely due to altered myogenesis caused by decreased Notch signaling resulting from the D233E mutation. Indeed, a recent report showed that α-dystroglycan glycosylation is at a low level during the first stages of muscle regeneration even in wild-type mice and is increased as myogenesis proceeds (Goddeeris *et al*, 2013). The authors proposed that coordination between myogenic differentiation and increased expression of the glycosyltransferase LARGE results in bulkier α-dystroglycan molecules with more extensive LARGE-mediated glycosylation and thereby a more robust ligand-binding capacity during muscle repair (Goddeeris *et al*, 2013). Accordingly, α-dystroglycan hypoglycosylation in our patients can be a consequence of altered SC homeostasis and differentiation, which would impede muscle repair and potentially result in immature LARGE-mediated glycosylation (Fig 9). Previous data show that an adequate balance of Notch activation is needed during myogenesis (Kuang *et al*, 2007; Sacco *et al*, 2008), and our results support an additional critical role of Notch signaling in the process of α-dystroglycan glycosylation during muscle differentiation. Nonetheless, the muscle-specific α-dystroglycan hypoglycosylation could contribute to muscle dystrophy in our patients.

An autosomal-dominant dermatosis has recently been associated with heterozygous mutations in *POFUT1* (protein *O*-fucosyltransferase

1, which adds *O*-fucose to Notch), and *POGLUT1* (Li *et al*, 2013; Basmanav *et al*, 2014). *Drosophila* Pofut1 has a chaperone-like function on the Notch protein independent of its *O*-fucosyltransferase activity (Okajima *et al*, 2005). Therefore, since patients homozygous for D233E have deficient enzymatic activity but normal POGLUT1 expression and no skin abnormalities, this dermatosis could be due to a defect in a non-enzymatic function of POGLUT1. However, genetic background differences could also exist that make these patients more sensitive to a 50% reduction in POGLUT1 enzymatic activity.

In conclusion, we identified the human *POGLUT1* D233E mutation that impairs Notch posttranslational modification and maintenance of muscle stem cells and leads to muscular degeneration and α-dystroglycan hypoglycosylation. Our findings demonstrate that D233E mutation in POGLUT1 causes autosomal recessive limb-girdle muscular dystrophy and implicate a primary defect in muscle progenitor cells as a novel pathomechanism for muscular dystrophy. Our patients display an adult-onset pure muscular phenotype, in contrast to the lethal, multisystemic phenotypes observed in animals with complete loss of POGLUT1 (Acar *et al*, 2008; Fernandez-Valdivia *et al*, 2011). Thus, the residual *O*-glucosyltransferase activity in our patients is sufficient for other aspects of human development and tissue homeostasis, indicating that myoblasts are particularly sensitive to decreased POGLUT1 activity.

# Materials and Methods

### Skeletal muscle histology and ultrastructural study

Open muscle biopsies were obtained from biceps brachii or quadriceps of the four patients and the healthy sibling in generation II of the family. The muscle was rapidly frozen in liquid nitrogen-chilled isopentane and 7-μm-thick cryostat sections were stained for standard histological and histochemical techniques including hematoxylin and eosin, Gomori trichrome, nicotinamide adenine dinucleotide dehydrogenase (NADH), and ATPase pH 9.4. Stained sections were evaluated with an Olympus BX41 (Tokyo, Japan) equipped with a ColorView IIIu camera (Olympus).

A piece of muscle from each biopsy was fixed in glutaraldehyde, embedded in Embed 812 resin, and processed for electron microscopy by standard methods (Malfatti *et al*, 2013). Ultrathin sections were viewed with Philips CM10 electron microscope (Eindhoven, Netherlands) equipped with Gatan 1k CCD camera.

### Antibodies

The list of the primary antibodies used in this study is provided in Appendix Table S7. The anti-α-dystroglycan core protein (a kind donation from S. Kröger) sheep polyclonal antibody was raised against a synthetic peptide corresponding to the carboxy-terminal 20 amino acids of chick α-dystroglycan (Herrmann *et al*, 2000).

### Immunohistochemical studies

Unfixed frozen muscle sections were incubated with primary antibodies overnight at 4°C, and then with the appropriate secondary antibodies for 30 min.

Cultured myoblasts were permeabilized in 0.2% Triton X-100 for 10 min and incubated in 1% BSA/PBS for 45 min; frozen sections were incubated in 5% normal goat serum/PBS for 45 min. Primary antibodies were used as described in Appendix Table S7 and incubated for 12 h at 4°C. After washing, samples were incubated with the appropriate secondary antibodies for 1 h.

PAX7 immunohistochemistry was carried out according to an antigen retrieval protocol, modified from Song *et al* (2012). Briefly, samples were fixed with 4% paraformaldehyde, pH = 7.4 (PFA) for 15 min. Sections were then immersed in citrate buffer (pH 6) for 30 min at 80°C and washed with PBS. Afterward, tissue samples were blocked with non-fat milk at 2% in PBS for 30 min and washed with PBS. Next, tissue sample were incubated with primary antibody for 2 days and secondary antibody for 2 h in blocking solution (5% BSA, 0.5% Triton X-100 in PBS). αDG-CORE immunohistochemistry required incubation with both biotinylated secondary antibody for 30 min and streptavidin conjugated to Cy3 for 15 min in PBS. Finally, the nuclei were stained for 20 min with To-pro-3-Iodide (Topro) at 1:1,000 in PBS and the slides were cover-slipped with fluorescence mounting medium (Dako). The images were acquired on a Zeiss LSM 710 confocal laser scanning microscope. Maximal projections of Z-stacked images were obtained and analyzed with ImageJ software.

### Glycoprotein enrichment and Western blot analysis

Frozen muscle samples and fresh skin fibroblasts were homogenized in RIPA buffer (20 mM Tris–HCl pH 7.4, 150 mM NaCl, 1 mM EDTA, 1% IGEPAL, 0.1% SDS) containing protease inhibitor mixture. Western blots of glycoproteins were enriched with wheat-germ agglutinin (WGA) agarose (Sigma-Aldrich) as described previously (Michele *et al*, 2002). Equivalent amounts of protein lysates non-incubated or incubated with WGA agarose beads were resolved on 7–10% SDS–PAGE gels and transferred to PDVF membranes (Millipore). Immunoreactivity was detected with secondary antibodies conjugated to horseradish peroxidase (Jackson Immuno Research) and developed with SuperSignal West Femto (Thermo Scientific) using an ImageQuant LAS 4000 MiniGold System (GE Healthcare).

### Ligand overlay assay

Ligand overlay assays in muscle and skin fibroblasts were performed as previously described with minor modifications (Michele *et al*, 2002; Willer *et al*, 2012). Briefly, PVDF membranes were incubated with Engelbreth-Holm-Swarm laminin (Sigma-Aldrich) or agrin (R&D Systems), overnight at 4°C in laminin-binding buffer. Then, membranes were washed and incubated with anti-agrin (Millipore) or anti-laminin (Sigma-Aldrich) primary antibodies and their corresponding secondary antibodies. Blots were imaged using the protocol described for Western blots.

### Flow cytometry

The following method was modified from Stevens *et al* (2013b). Briefly, an anti-α-dystroglycan antibody IIH6 was used to assess the amount of α-dystroglycan glycosylation in fibroblasts from patients (II.4 and II.5) and two controls. Fibroblasts (passages 2–3) were

grown until approximately 90% confluent. Cells were detached using Accutase (Sigma, UK), centrifuged for 3 min at 500 *g*, and counted and resuspended in PBS to a final density of 200,000 fibroblasts per milliliter, centrifuged at 3,000 *g* for 3 min, and incubated on ice with IIH6 for 30 min, anti-mouse biotinylated IgM (Vector Labs, USA) for 20 min, and streptavidin-PE (BD Pharmingen, UK) for 15 min. After washing, cells were resuspended in 500 ml of PBS and transferred to FACS tubes (Corning Science, Mex). Data were acquired using the BD FACS Canto II analyzer and analyzed using the BD FACSDiva software (BD Bioscience, US). Two separate controls for each fibroblast population were used: one without any staining to remove background and another without the primary antibody to gate the IIH6-positive population. A total of 50,000 cells were analyzed per experiment. We assessed the percentage of IIH6-positive cells as well as the level of IIH6 the mean fluorescence intensity (MFI).

### Genetic mapping

Genomewide genotyping using the Illumina Infinium HumanOmni1-Quad, v1.0 BeadChip (Illumina, San Diego, CA, USA) was performed in the DNA samples of the four affected siblings (II.1, II.2, II.4, and II.5) and the unaffected sister (II.3), thus allowing the analysis of over one million assays for each sample. The chips were scanned, and data were loaded into Illumina Genome Studio V2009.1 software. Genome Viewer tool was used to visualize genomic copy number variation as well as individual genotypes. Log R ratio and B allele frequency, plotted along the entire genome for all SNPs on the array, were used to analyze all variants. CNV partition 2.4.4 was employed to automatically detect copy number variants as well as regions of extended homozygosity > 500 kb across the genome. Genotype success rates > 99.7% were obtained for all assays (Appendix Fig S6).

### Exome sequencing

The capture and subsequent sequencing of exonic regions of DNA samples II.3, II.4, and II.5 was performed at Otogenetics Co (Norcross, GA, USA) using the solution-based SeqCap EZ Human Exome Library v2.0 (Roche NimbleGen, Madison, WI, USA) and the Illumina HiSeq2000, with 100 paired-end runs. An average coverage of 30× across the exome was achieved for all samples. Reads were aligned to the reference genome hg19 with DNAnexus software (Palo Alto, CA, USA) using default parameters. Single nucleotide polymorphisms and indels were identified and realigned using the Genome Analysis Toolkit (GATK). In order to discard common variants, genetic variations were further verified in the Database of Single Nucleotide Polymorphisms (dbSNP Build ID: 137; http://www.ncbi.nlm.nih.gov/SNP/), the exome variant server (EVS) of the National Heart, Lung, and Blood Institute GO Exome Sequencing Project (Seattle, WA, USA; http://evs.gs.washington.edu/EVS/), and the 1000 Genomes Project (http://www.1000genomes.org/).

### Segregation analysis

Segregation analysis was performed by means of Sanger sequencing using the primers presented in Appendix Table S4. Cycle sequencing

was performed using the Dye Terminator Sequencing Kit (Applied Biosystems, Foster City, CA) and run on an ABI 3100 genetic analyzer. Sequence chromatograms were analyzed using the Sequencher software (Genecodes, Ann Arbor, MI).

### Genotyping

Genotyping of the p.D233E mutation in *POGLUT1* gene in the Spanish and the Human Genome Diversity Project series was carried out by real-time PCR using a custom TaqMan® Genomic Assay (Applied Biosystems, Foster City, CA) with appropriate primers and probes (Appendix Table S5). Thermal cycling and end-point polymerase chain reaction (PCR) analysis was performed on a real-time PCR instrument (ABI 7900HT, Applied Biosystems, Foster City, CA). DNA from family members II.3 and II.4 was used as positive controls in all experiments.

### *POGLUT1* subcloning and mutagenesis

Human *POGLUT1* cDNA cloned into the pEZ-M02 vector was obtained from GeneCopoeia (Rockville, MD, USA). The p.D233E mutation was introduced with the QuikChange II site-directed mutagenesis kit (Agilent Technologies, Santa Clara, CA, USA), as per manufacturer's indications, and confirmed by direct sequencing. The oligonucleotide sequences used for site-directed mutagenesis and Sanger sequencing were as follows: POGLUT1 5′-CAGCCTCCG GACTCTAGC-3′ and 5′-TAATACGACTCACTATAGGG-3′ for wild-type and POGLUT1 5′-TCTCGGAAAAACCCAAAACTTGTTGA<u>G</u>GC AGAATACACCAAA-3′ and 5′-TTTGGTGTATTCTGC<u>C</u>TCAACAAGTT TTGGGTTTTTCCGAGA-3′ (mismatched nucleotide to the reference sequence is underlined) for the D233E mutant. cDNAs encoding the luminal domains of wild type and D233E mutant of human POGLUT1, starting from Arg24, were amplified by PCR using the primers, 5′-ATATATAAGCTTCACGCCAGAAGGAGTCAGGTT-3′ and 5′-ATCTAGCTCGAGCTAGTTCAGTTTTCAACATTTTGGG-3′, and subcloned into a pSecTag2c vector (Invitrogen) in frame with a C-terminal Myc/6xHis tag for expression in mammalian cells. The expression constructs for wild-type human POGLUT1 in pcDNA4 or pTracer vector were also used as a template for site-directed mutagenesis to introduce the D233E mutation (Rana *et al*, 2011; Takeuchi *et al*, 2012). The sequences of the primers are 5′-CT TGTTGAGGCAGAATACACCAAAAACCAGGCCTG-3′ and 5′-GTATT CTGCCTCAACAAGTTTGGGTTTTTCCGAGACAG-3′. The pcDNA4-based construct encodes the recombinant protein with a C-terminal Myc/6xHis tag. The pTracer-based construct encodes the recombinant protein with a FLAG tag, which is immediately downstream of the KTEL sequence at the C-terminus. Successful incorporation of the mutation was confirmed by direct DNA sequencing.

### *In vitro* glycosylation assays

Myc/6xHis-tagged POGLUT1 proteins were expressed in HEK293T cells by transient transfection, and the secreted proteins were purified from the culture media using Ni-NTA agarose (QIAGEN). POGLUT/POXYLT assays were performed as previously described (Fernandez-Valdivia *et al*, 2011). Briefly, a 10 μl standard reaction mixture contained 50 mM HEPES pH 6.8, 10 mM $MnCl_2$, the indicated amounts of acceptor substrates, 0.16 μM UDP-[6-$^3$H]glucose

(2.22 TBq/mmol), 10 μM UDP-glucose, 0.5% Nonidet P-40, and purified proteins. For POXYLT assay, 10 μM UDP-[$^{14}$C(U)]xylose (5.43 GBq/mmol) was used as donor substrate. The reaction was performed at 37°C for 20 min and stopped by adding 900 μl of 100 mM EDTA pH 8.0. The sample was loaded onto a C18 cartridge (100 mg, Agilent Technologies). After the cartridge was washed with 5 ml of $H_2O$, the EGF repeat was eluted with 1 ml of 80% methanol. Incorporation of [6-$^3$H]glucose or [$^{14}$C(U)]xylose into the EGF repeats was determined by scintillation counting of the eluate. Reactions without substrates were used as background control. Data are from three independent assays. The values indicate mean ± SEM.

*O*-glucosylation or *O*-xylosylation of human factor IX EGF repeat and EGF16 from mouse Notch2 was performed using a modified POGLUT/POXYLT assay. A 100 μl reaction mixture contained 50 mM HEPES pH 6.8, 10 mM $MnCl_2$, 10 μM acceptor substrate, 200 μM UDP-glucose, and purified wild-type or D233E mutant POGLUT1. For *O*-xylosylation, 200 μM UDP-xylose was used as donor substrate. Approximately 500 ng of each recombinant enzyme was used. The reaction was performed at 37°C overnight. The reaction products were purified by reverse-phase high-performance liquid chromatography (RP-HPLC, Agilent Technologies, 1200 Series) equipped with a C18 column (10 × 250 mm, VYDAC) with a linear gradient of solvent B (80% acetonitrile, 0.1% trifluoroacetic acid (TFA) in water) from 10 to 90% in solvent A (0.1% TFA in water) for 60 min, monitoring absorbance at 214 nm. Elution profiles between 20 and 30 min are shown as milli-absorbance unit (mAU). Peak fractions were pooled, dried down in a Speed-Vac centrifuge, and stored at −20°C for further mass spectral analysis.

## Mass spectrometric analysis of glycosylation products

Mass spectrometric analysis by infusion was performed as described previously (Takeuchi *et al*, 2012). Briefly, dried samples were resuspended in 0.1% formic acid in water, spin-filtered, and injected into an Agilent 6340 ion-trap mass spectrometer with a nano-HPLC CHIP–Cube interface at a rate of 18 μl/h. The MS peaks for MS/MS were chosen manually, and the data were analyzed using Agilent ChemStation data analysis software. The masses of EGF repeats with different charge states were deconvoluted and shown on the top of peaks in HPLC profiles.

## RT–PCR and qRT–PCR analysis

Total RNA was extracted using QIAzol and RNeasy kit (Qiagen). cDNA was synthesized from 1 μg of total RNA using Transcriptor First Strand cDNA Synthesis Kit (Roche). qRT–PCR (100 ng total cDNA per reaction) was performed using TaqMan® Fast Advanced Master Mix (Applied Biosystems) in a 7500 Fast Real-Time PCR System. Relative mRNA levels were compared using the $2^{-\Delta\Delta C_T}$ method, with *GAPDH* as control (only POGLUT1 mRNA level was relative to desmin). We used TaqMan probes (Applied Biosystems) for human *HES1* (Hs00172878_m1), human *PAX7* (Hs0024 2962_m1), human *POGLUT1* (Hs00220308_m1), human *GAPDH* (Hs02758991_g1), human *DES* (Hs00157258_m1), mouse *HES1* (Mm01342805_m1), mouse *Pax7* (Mm01354484_m1), and mouse *GAPDH* (Mm03302249_g1).

We selected the disease control muscles from patients with well-characterized limb-girdle muscular dystrophies. To avoid possible

differences due to the effect of age on the satellite cell population (Chakkalakal *et al*, 2012; Sousa-Victor *et al*, 2014), our samples of D233E patients and both healthy and disease controls were age-matched (Appendix Table S6).

## Generation of the transgenic fly

To generate transgenic flies capable of overexpressing human POGLUT1$^{D233E}$, a BamHI-XbaI fragment containing the POGLUT1$^{D233E}$ ORF was released from pTracer-POGLUT1$^{D233E}$-FLAG and cloned into the BglII-XbaI sites of *pUASTattB* (Bischof *et al*, 2007) to generate pUASTattB-POGLUT1$^{D233E}$-FLAG. Generation of pUASTattB-POGLUT1$^{wt}$-FLAG was described previously (Fernandez-Valdivia *et al*, 2011). Both constructs were integrated into the *VK22* docking site by using ΦC31-mediated transgenesis to generate *UASattB-POGLUT1*$^{D233E}$*-FLAG.VK22* and *UASattB-POGLUT1*$^{WT}$*-FLAG.VK22* transgenes. Correct integration events were verified by att PCRs, as described previously (Venken *et al*, 2006).

## Analysis of muscle phenotype in *Drosophila*

We assessed the effect of POGLUT1 activity toward the development of indirect flight muscles in *Drosophila* (Gildor *et al*, 2012). Rescue experiments were performed with fly transgenes expressing human *POGLUT1*$^{wt}$ (wild type) and human *POGLUT1*$^{D233E}$. *w*$^{1118}$ (wt), *rumi*$^{79/79}$, *Mef2-GAL4/UASattB-POGLUT1*$^{WT}$*-FLAG; rumi*$^{79/79}$, and *Mef2-GAL4/UASattB-POGLUT*$^{D233E}$*-FLAG; rumi*$^{79/79}$ animals were raised at 18°C until puparium formation. Pupae 0–1 h after puparium formation (APF) were incubated at indicated temperatures until 25% pupal development. Pupal indirect flight muscles were dissected and stained using standard methods. Antibodies are rabbit α-Twist 1:5,000 (Roth *et al*, 1989), mouse 22C10 1:50 (DSHB), mouse α-FLAG M2 1:100 (Sigma), donkey-α-rabbit-Cy3 1:500, donkey-α-mouse-Cy5 1:500 (Jackson ImmunoResearch Laboratories). Confocal images were scanned using a Leica TCS-SP5 microscope and processed with Amira5.2.2. Myotube lengths were measured using a two-dimensional measurement tool in Amira5.2.2. Mean myotube lengths and standard deviations were calculated for *rumi*$^{79/79}$ pupae with no rescue and *rumi*$^{79/79}$ pupae overexpressing *POGLUT1*$^{WT}$ ($n = 22$) or *POGLUT1*$^{D233E}$ ($n = 54$). One-way ANOVA with Tukey's multiple comparisons test was used to determine the *P*-values. Images were processed with Adobe Photoshop CS5; figures were assembled in Adobe Illustrator CS5.

## C2C12 cell line culture and Notch inhibition assay

C2C12 cell line was cultured in monolayer. The culture media for the C2C12 myoblast stage contains Dulbecco's minimal essential medium (DMEM, Gibco, BRL, Bethesda, MD), supplemented with 5% glutamine, 10% fetal bovine serum, 1% penicillin–streptomycin–fungizone, and 23 mM HEPES. The cultures were examined regularly to avoid overgrowth. For the Notch inhibition assay, we treated C2C12 myoblasts at 80–90% confluence with DAPT (Sigma-Aldrich, USA), a γ-secretase inhibitor, and a well-established inhibitor of Notch, a γ-secretase substrate, as previously published (Kuang *et al*, 2007). We also treated C2C12 myoblasts with LY3039478 (ApexBio Technology LLC), a more specific and potent Notch inhibitor. For this experiment, culture proliferation and

differentiation medium were modified by adding 50 μM DAPT or 25 μM LY3039478, and 50–25 μM DMSO for controls, respectively.

### Primary cell culture

Muscle biopsies from two patients (II.4 and II.5), three age-matched healthy controls and three age-matched disease controls were used for the experiments. Muscle biopsies were minced and cultured in a monolayer. Briefly, the culture medium for the myoblast stage contains 75% Dulbecco's minimal essential medium (Invitrogen) and 25% M199 medium (Invitrogen), supplemented with 10% fetal bovine serum (FBS), 10 μg/ml insulin, 2 mM glutamine, 100 units/ml penicillin, 100 μg/ml streptomycin, 0.25 μg/ml fungizone, 10 ng/ml epidermal growth factor, and 25 ng/ml fibroblast growth factor. To obtain highly purified myoblasts, each $10^7$ cells were mixed with 20 μl of CD56-coated microbeads (Miltenyi Biotec, Bergisch Gladbach, Germany) and incubated at 6–12°C for 15 min. Unbound microbeads were removed by washing cells in excess PBS buffer followed by centrifugation at $300 \times g$ for 10 min. The cell pellet was resuspended in PBS buffer to a concentration of $2 \times 10^8$ cells/ml before separation on a midiMACS cell separator (Miltenyi Biotec, Bergisch Gladbach, Germany). CD56-positive cells were seeded at 13,000 cells/cm$^2$ using the culture medium for the myoblast stage containing 15% of FBS and without growth factors to avoid interference with the results. We examine proliferation at different days. When the myoblasts started to fuse, the medium was substituted with one containing 2% of FBS. We measured the fusion index 4 days after medium change by calculating the mean percentage of nuclei in myotubes in respect to the total number of nuclei (myoblasts + myotubes).

Human skin biopsies were minced and cultured. Fibroblasts were grown in DMEM containing 15% fetal bovine serum and 1% penicillin–streptomycin–fungizone at 37°C with 5% $CO_2$.

### Myoblast immortalization

Myoblasts from II.4 patient and healthy control were transduced with lentiviral vectors encoding hTERT and cdk4 containing puromycin and neomycin selection markers, respectively. Transduced cells were selected with puromycin (1 μg/ml) for 6 days and neomycin (1 mg/ml) for 10 days. Infected cells were purified using an immunomagnetic cell sorting system (MACS; Miltenyi) according to the manufacturer's instructions with anti-CD56 microbeads. Cells were seeded at clonal density, and individual myogenic clones were isolated and characterized.

### Fibroblast myogenic conversion

Fibroblasts from II.4 patient and healthy control were transduced with lentiviral vector encoding hTert. Transduced cells were selected with puromycin (1 μg/ml) for 6 days. Immortalized fibroblasts were then transduced with a lentiviral vector encoding inducible MyoD. For MyoD induction, doxycycline (2 μg/ml) was added in the proliferation medium (this day was considered day 0).

### Lentivirus generation and infection

We generated lentiviral vectors (LV), based on the calcium phosphate method as previously described (Yoshida *et al*, 2013). Briefly,

HEK293T cells were transfected using transfer plasmids, EF.hICN1.CMV.GFP (#17623, Addgene) or EF.CMV.GFP, together with psPAX2 packaging and pMD2.G envelope plasmid DNA at a ratio of 4:3:1, respectively. Then, 48 h post-transfection, medium was collected, filtered, and concentrated by centrifugation for 90 min at $105,000 \times g$ using a SW 32 Ti Beckman rotor. Supernatant was completely removed and pellet diluted in 100 μl PBS overnight at 4°C, titered by FACS, and stored at −80°C until use. Immortalized myoblasts from patients were infected with LV-NICD-GFP or LV-GFP, and immortalized myoblasts from control with LV-GFP. After 7 days, GFP-positive cells were purified by FACS.

### Study approval

This study was approved by the Institutional Research Ethic Committee at Hospital Universitario Virgen del Rocío in Sevilla (Spain). Written informed consent was received from participants, prior to inclusion in the study, for genetic studies, for muscle and skin biopsies, and for pictures appearing in the manuscript.

**Expanded View** for this article is available online.

### Acknowledgements

We thank Isabel Illa (Hospital Santa Creu i Sant Pau) and Salvatore DiMauro (Columbia University) for their critical comments; Montserrat Olivé (Hospital de Bellvitge) for her supporting on muscle imaging; Cristina Dominguez (Hospital 12 Octubre) for ceding us muscle samples; and Gloria Cantero for helping with quantifications. Stephan Kröger (Münich University) for kind donation of the antibody against the α-dystroglycan core (clone no. 317); Maria Leptin (Cologne University) for anti-Twist antibody; David Comas (Institut de Biología Evolutiva, Pompeu Fabra University), for providing DNA samples; Developmental Studies Hybridoma Bank for 22C10 antibody; Bloomington *Drosophila* stock center (NIH P40OD018537) for the *Mef2-GAL4* strain; Confocal Microscopy Core of the BCM IDDRC (1U54HD083092; the Eunice Kennedy NICHD); Platform for immortalization of human cells of the Myology Institute in Paris; and Raquel Gómez (Biomedicine Institute of Seville Pathology Core Facility), Juan J. Pérez-Moreno (Centro Andaluz de Biología del Desarrollo), and O. Akman (Columbia University) for their technical support. Supported in part by grants from the Health Institute Carlos III and FEDER (FIS PI10/02410, PI13-01739, and BA12-00097 to C. Paradas, FIS12/2291 to E. Gallardo, and CIBERNED to R. Fernández-Chacón), the Andalusian Government (PI-0017-2014 to C. Paradas and P12-CTS-2232 to R. Fernández-Chacón), The Matsumae International Foundation (to C. Paradas), the NIH/NIGMS (R01GM061126 to R.S. Haltiwanger and R01GM084135 to H. Jafar-Nejad), the NIH (U54 NS078059 from NINDS and NICHD, R01 HD056103 from NICHD to M. Hirano), and the MDA USA (to M. Hirano).

### Author contributions

CP and JC designed the study. ES-M performed α-dystroglycan expression and function studies, cell culture, satellite cells, and myogenesis analysis. HT performed biochemical *in vitro* assays. TVL and BE performed the fly studies. CP, MC-S, YM, and CM handled patients and collected biological samples from the patients. ER and ES-M processed and studied muscle biopsy. CP, MC-S, and JAM-L analyzed the radiological findings. FM, CP, JLN-G, and LG-S analyzed protein expression and mRNA levels. JC and GP performed and genetic studies. MCR and XS-C performed cell culture. AB generated the immortalized cells. EA-G, EG, and RF-C provided critical discussion on the

<div style="background:#e8f0f7;padding:1em;">

**The paper explained**

**Problem**

The fact that in a significant number of patients suffering from a muscular dystrophy, the responsible gene is still unknown stimulates research to describe new molecular and cellular pathomechanisms underlying muscle degeneration. It is known that even in healthy individuals, common daily activities cause skeletal muscle lesions that are physiologically repaired through activation of satellite cells. Accordingly, it is logical to envision a scenario where alteration in the satellite cell homeostasis could lead to muscle fiber degeneration because of accumulation of unrepaired damage. Indeed, this appears to be the case with the family we present carrying a homozygous missense D233E mutation in *POGLUT1* gene: a muscular dystrophy as a consequence of defective muscle regeneration.

**Results**

It is widely known that Notch signaling pathway plays critical roles in the highly coordinated muscle regenerative process, maintaining an appropriate population of satellite cells and preventing premature differentiation. Our previous work in *Drosophila* and mammalian cell lines has shown that the addition of *O*-linked glucose to Notch receptors by protein *O*-glucosyltransferase 1 (POGLUT1; also known as Rumi) is required for Notch signaling. In the investigated family, a homozygous missense D233E mutation in *POGLUT1* dramatically reduces its enzymatic activity on Notch. As a result, the investigated family shows a defect in Notch signaling with a significant depletion of satellite cells, resulting in defective muscle regeneration and ultimately leading to muscle dystrophy. In addition, primary myoblasts from our patients displayed reduced self-renewal capacity and enhanced differentiation in culture, as expected under a Notch signaling defective environment. We show that the enzymatic activity of POGLUT1 is also essential for the formation of adult muscles by myoblasts in *Drosophila*. Moreover, cross-species overexpression studies in *Drosophila* indicate that the D233E mutation impairs the ability of human POGLUT1 in rescuing the muscle defects in flies lacking endogenous POGLUT1 activity. We have previously shown that complete loss of POGLUT1 results in embryonic lethality, but our patients only exhibit an adult-onset muscular dystrophy. Our data provide a clear explanation for this apparent discrepancy: First, the D233E does not affect the expression level and the subcellular localization of POGLUT1, and second, despite the severe decrease in the enzymatic activity of POGLUT1$^{D233E}$, this mutant protein maintains a residual activity. Of note, our patients also exhibited a muscle-specific reduction in glycosylation and laminin binding of α-dystroglycan. While these defects probably contribute to muscle dystrophy in our patients, we do not favor the hypothesis that they constitute the primary underlying mechanism for the disease, as α-dystroglycan lacks POGLUT1 target sites and patient fibroblasts show normal α-dystroglycan glycosylation and laminin binding. Instead, we propose that altered proliferation and differentiation of satellite cells due to decreased Notch signaling ultimately alter the α-dystroglycan glycosylation, which is a dynamic process during muscle differentiation.

**Impact**

Our data link a missense mutation in *POGLUT1* to a novel form of muscular dystrophy and suggest that Notch-dependent loss of satellite cells is the primary pathomechanism for the disease in our patients. The muscle-specific phenotypes in our patients indicate that muscle stem cells are much more sensitive to a decrease in POGLUT1 enzymatic activity compared with other human cell types and suggest that small molecules affecting the activity of POGLUT1 in muscle could be of potential benefit in controlling the behavior of satellite cells, paving the way for potential future therapies.

</div>

research. CP, MH, RSH, and HJ-N supervised and mentored all work. CP coordinated all the study and wrote the initial manuscript. All authors contributed to the final version of the manuscript.

## Conflict of interest

The authors declare that they have no conflict of interest.

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
