## [Review Process File · EMBO Molecular Medicine]

A *POGLUT1* mutation causes a muscular dystrophy with reduced Notch signaling and satellite cell loss

Emilia Servián-Morilla, Hideyuki Takeuchi, Tom V Lee, Jordi Clarimon, Fabiola Mavillard, Estela Area-Gómez, Eloy Rivas, Jose L Nieto-González, Maria C Rivero, Macarena Cabrera-Serrano, Leonardo Gómez-Sánchez, Jose A Martínez-López, Beatriz Estrada, Celedonio Márquez, Yolanda Morgado, Xavier Suárez-Calvet, Guillermo Pita, Anne Bigot, Eduard Gallardo, Rafael Fernández-Chacón, Michio Hirano, Robert S Haltiwanger, Hamed Jafar-Nejad, Carmen Paradas

Corresponding author: Carmen Paradas, Instituto de Biomedicina de Sevilla/Hospital Universitario Virgen del Rocío

Review timeline:

Submission date:	08 September 2015
Editorial Decision:	13 October 2015
Revision received:	09 April 2016
Editorial Decision:	30 May 2016
Revision received:	16 July 2016
Editorial Decision:	08 August 2016
Revision received:	22 August 2016
Accepted:	02 September 2016

Transaction Report:

Editor: Roberto Buccione

1st Editorial Decision

13 October 2015

Thank you for the submission of your manuscript to EMBO Molecular Medicine. We have now heard back from the Reviewers whom we asked to evaluate your manuscript. We are sorry that it has taken longer than usual to get back to you on your manuscript. In this case we experienced unusual difficulties in securing appropriate reviewers and further to this, one Reviewer withdrew suddenly and unexpectedly.

As you will see the issues raised are few but fundamental. Although I will not dwell into much detail, I would like to highlight the main points.

The main item of concern is that the data do not convincingly support the mechanism proposed, specifically that Notch down-regulation causes the dystrophic phenotype. In part connected to this is the concern expressed by both reviewers that data derive from only one patient and control.

Reviewers 1 also notes that the muscle biopsies are not clearly showing a dystrophic pattern and that

many experimental details are lacking. S/he also lists other items that require action.

Reviewer 2 is also reserved and suggests a number of approaches to improve the experimental foundation of the manuscript, including testing specific points on mouse material. S/he also notes, as does Reviewer 1, the unconvincing data on muscle dystrophy. Reviewer 2 provides a detailed list of other critical points.

In conclusion, while publication of the paper cannot be considered at this stage, given the potential interest of your findings and after further discussion with my colleagues and reviewer cross-commenting, we have decided to give you the opportunity to address the above concerns.

We are thus prepared to consider a substantially revised submission, with the understanding that the Reviewers' concerns must be addressed with additional experimental data where appropriate and that acceptance of the manuscript will entail a second round of review. This includes providing convincing causal evidence for Notch implication. The overall aim is to significantly upgrade the clinical relevance and usefulness of the dataset, which of course is of paramount importance for our title.

I understand that if you do not have the required data available at least in part, to address the above, this might entail a significant amount of time, additional work and experimentation and might be technically challenging, I would therefore understand if you chose to rather seek publication elsewhere at this stage. Should you do so, we would welcome a message to this effect.

It is important that you consider that it is EMBO Molecular Medicine policy to allow a single round of revision only and that, therefore, acceptance or rejection of the manuscript will depend on the completeness of your responses included in the next, final version of the manuscript.

As you know, EMBO Molecular Medicine has a "scooping protection" policy, whereby similar findings that are published by others during review or revision are not a criterion for rejection. However, I do ask you to get in touch with us after three months if you have not completed your revision, to update us on the status. Please also contact us as soon as possible if similar work is published elsewhere.

Finally, please note that EMBO Molecular Medicine now requires a complete author checklist (<http://embomolmed.embopress.org/authorguide#editorial3>) to be submitted with all revised manuscripts. Provision of the author checklist is mandatory at revision stage; The checklist is designed to enhance and standardize reporting of key information in research papers and to support reanalysis and repetition of experiments by the community. The list covers key information for figure panels and captions and focuses on statistics, the reporting of reagents, animal models and human subject-derived data, as well as guidance to optimise data accessibility.

***** Reviewer's comments *****

Referee #1 (Comments on Novelty/Model System):

I am unconvinced by the tissue culture data because there are insufficient numbers (see below). The fly model is useful but not conclusive since there are differences with respect to POGLUT1 between fly and human and it is a partial rescue.

Referee #1 (Remarks):

This paper is very well written and is very clear. The figures are of excellent quality and allow for critical analysis, for this the authors should be applauded. However, the main issues with this paper are that it is based upon data from one family - at least another should be included for verification of the findings and that the tissue culture experiments are based upon one patient and one control. Myoblast cultures are extremely variable and there needed to be not only more patient cultures and aged matched controls but also disease controls to verify the authors hypothesis that Notch signalling is disrupted. It may very well be proven that this is the case but the evidence presented

here is insufficient to justify statements in the abstract that link the mutation and reduced activity of POGlut1 to impaired muscle development, decrease in Notch signalling, dramatic reduction of the satellite pool etc. More specific comments are as follows:-

The data from the muscle biopsies do not clearly show a dystrophic pattern as stated in the text.

I was unclear as to whether POGlut1 may act on other proteins with EGF like repeats and thus influence the clinical phenotype.

The difference between laminin and agrin overlay assays is odd and deserves some discussion as to why this might be so.

Methodology is perhaps not as detailed as it might be in places - i.e. electron microscopy - what resin were the specimens embedded in, fine to say standard methods but there are many and also there is no reference,

the antibody to the core protein from Stephan Kroger - what species is this from - is it a monoclonal - a clone name is mentioned - this would help the data to be compared by the readers with other data obtained in dystroglycanopathy patients.

Pax7 immunolabelling should always be carried out with a basement membrane marker to ensure the correct location of the staining.

Minor comment alpha dystroglycan glycosylation is not decreased during the first stages of muscle regeneration it is at a low level which as the authors subsequently state increases. This is different to a decrease.

In conclusion - the hypotheses underlying this investigation are interesting but the data does not prove or disprove them.

Referee #2 (Comments on Novelty/Model System):

The data full significantly short of supporting the authors conclusions. The manuscript is not suitable for publication.

Referee #2 (Remarks):

In this paper Servian-Morilla and colleagues identified a family of dystrophic patients with a missense mutation in POGlut1. The authors suggest that lack of POGlut1 enzymatic activity reduces Notch activity, which results in impaired satellite cell function and reduced -DAG glycosylation. The paper contains significant amount of data, however in its present form, the evidence presented in that paper do not persuasively support the mechanism proposed by the authors. Especially, the authors do not convincingly demonstrate that Notch downregulation is directly causing the dystrophic phenotype that they observe. Additional experiments are needed to support their claim.

Major comments:

- The authors suggest that Notch is the primary cause of the dystrophic phenotype they observed, although this is only indirectly addressed. Experiment on drosophila partially address that point but it measures muscle formation, which is quite different than adult muscle. The authors need to perform a Notch overexpression experiment on their myoblasts isolated from the dystrophic patients to measure whether it rescues the satellite cell functions (proliferation, self-renewal, differentiation, fusion)

- Figure 5 shows decreased Notch activity and PAX7-expressing cells in the dystrophic patients. The authors suggest that reduced Notch activity leads to impaired satellite cell function, which ultimately results in dystrophic-like muscle phenotype. However, the opposite could also be true, e.g. the decreased Notch activity could be a consequence of the dystrophic phenotype and not the primary cause. For example, myofiber fragility caused by -DAG hypoglycosylation could induce muscle degeneration and regeneration, which would lead to satellite cell activation and reduced Notch activity. Alternatively, many proteins other than Notch have EGF-like repeats and could thus be affected by mutation in POGlut1.

Therefore, this is clearly speculative at this point whether the hypoglycosylation of -DAG is a

consequence of reduced Notch activity. This point needs to be addressed to clearly understand the mechanism. For example, siRNA of POGLUT1 on primary myotubes should help to determine whether POGLUT1 can act directly on -DAG glycosylation. Alternatively, The authors could measure whether a rescue of Notch in their primary myoblast model affects the level of -DAG glycosylation. If the mechanism proposed by the authors is correct, Notch overexpression should rescue -DAG glycosylation, otherwise these 2 processes are parallel, or -DAG glycosylation acts upstream of Notch.

- Figure 7 shows impaired myoblast function in the dystrophic patient. However, this experiment was done only on cells from one patient and one control. It is impossible to draw any conclusion from a single biological sample (even if technical replicates were done) because there is multiple other factors that could impact myoblast function other than the mutation in POGLUT1, such as age of the patient, gender, intrinsic variations, activation level of satellite cells when the biopsy was performed, etc. This experiment absolutely needs to be performed with biological triplicates both for the controls and the dystrophic patients. This reviewer understand the technical limitation of performing more biopsies on these patients (if it wasn't done previously), however, this figure is submitted to too many bias if it relies only on one patient, and this figure is critical to support the authors conclusions.

- POGLUT1 knockdown on isolated myofibers (from wildtype mice) would improve the author's conclusions regarding the effect of this protein on satellite cell function. The authors could then measure more directly and precisely the effect of POGLUT1 on satellite cell proliferation (total number of myogenic cells at 72h), differentiation (number of myogenin-positive cells at 72h), and self-renewal (number of pax7+/MyoD- cells at 72h).

- The authors argue that the decrease in Notch signaling and in PAX7-expressing cells in the D233E patients is not secondary to -DAG hypoglycosylation by comparing with other types of dystroglycanopathies. However, this might be reflecting the different levels of muscle damage. Only 1 out of 4 of the D233E patients present signs of degeneration (based on H&E staining in Fig 1) while 5 out of 8 disease controls patients have moderate to severe signs of dystrophy (based on suppl table S6). Higher levels of muscle degeneration will obviously lead to higher number of PAX7-positive cells. To draw any conclusion from this, the authors should also compare the patients based on intensity of the disease.

Specific points:

- Different samples from different patients were used for the various experiments. For example, patients 2 and 5 were used for figure 2a-b, but patient 3 and 4 were used for figure 2c. It is also unknown which patient was used for figure 7 (if it is patient 1 it could explain that the results are different than control since he has a more severe phenotype and his satellite cells were probably already activated when they were isolated). Anyhow, it should be explicitly stated which one, why and how each samples were chosen for each experiments.

- Isolation of satellite cells from human muscle samples could be challenging. Isolation with CD56 microbeads is a relatively good method, although contamination remains a concern. Myoblasts seem to proliferate more slowly in the D233E patient, thus a small fibroblasts contamination could take over the myoblast population after many days in culture (especially if the fibroblast are not affected by the mutation). Purity of myoblast population needs to be shown.

1st Revision - authors' response

09 April 2016

Referee #1 (Comments on Novelty/Model System):

I am unconvinced by the tissue culture data because there are insufficient numbers (see below).

The fly model is useful but not conclusive since there are differences with respect to POGLUT1 between fly and human and it is a partial rescue.

Referee #1 (Remarks):

This paper is very well written and is very clear. The figures are of excellent quality and allow for critical analysis, for this the authors should be applauded. However, the main issues with this paper are that it is based upon data from one family - at least another should be included for verification of the findings and that the tissue culture experiments are based upon one patient and one control. Myoblast cultures are extremely variable and there needed to be not only more patient cultures and aged matched controls but also disease controls to verify the authors hypothesis that Notch signaling is disrupted. It may very well be proven that this is the case but the evidence presented here is insufficient to justify statements in the abstract that link the mutation and reduced activity of POGLUT1 to impaired muscle development, decrease in Notch signaling, dramatic reduction of the satellite pool etc.

We have checked 50 DNA samples of our and other Spanish neuromuscular units, from muscular dystrophy patients without a genetic diagnosis. So far we have not found POGLUT1 mutations in any of them, so unfortunately we do not have more families to include in our study.

We and our local Ethic Committee have ethical concerns regarding trying to insistently convince patients II.1 and II.2 for a new biopsy (it would be the third and fourth biopsy for them). Thus, this was not an option. Thanks to a new muscle sample from patient II.4, we were able to add one more patient's sample for proliferation and differentiation assay. These myoblasts grow very slowly, likely because of the nature of the disease, and we want to clearly explain this point to state that, unfortunately, it was impossible to obtain enough primary myoblasts for rescue experiments. Fortunately, we were able to obtain 100.000 additional cells from this last sample to be immortalized. Thus, we have generated a continuous cell line from the patient's myoblasts and performed new proliferation and differentiation assays to confirm our previous results, and also performed the rescue experiments.

In summary, now we added:

- 1.1 One more proliferation assay (n=2), and the results confirm our previous conclusions (new data in Figure 7B-F).**
- 1.2 One more differentiation assay (n=2), and the results confirm our previous conclusions (new data in Figure 7G-I).**
- 1.3 We have obtained 6 more fresh muscle samples from 3 healthy controls and 3 dystrophic controls, so we have repeated the proliferation and differentiation experiments. Now we have n=3 for each condition and the results confirm our previous conclusions (new data in Figure 7A-F).**
- 1.4 We have repeated three independent proliferation and differentiation assays using the continuous cell lines created by immortalizing myoblasts from one patient and one healthy control; the results confirm our previous conclusions about the association between D233E mutation and slow proliferation and facilitated differentiation; new data in Supplementary Figure S11).**
- 1.5 In addition, since we had already fibroblasts isolated from skin biopsies from patients II.4 and II.5, we have immortalized them by transduction of hTERT and converted them to myogenic cells by transduction of inducible MyoD. We have performed proliferation and differentiation assays to confirm that these converted fibroblasts show similar phenotype to primary myoblasts (slow proliferation and facilitated differentiation; new data in Figure S12).**

In order to demonstrate that reduced activity of POGLUT1 impairs muscle development through decrease in Notch signaling we performed rescue experiments on D233E immortalized myoblasts by lentiviral overexpression of NICD. Increasing Notch signaling fully

rescued the proliferation defects in D233E myoblasts, which showed the same proliferation rate as controls. NICD overexpression during differentiation not only rescued the phenotype but also reversed the effect, as myotubes appeared later in D233E culture than in control cells without overexpression of NICD (control-LV-GFP). This result supports the dominant pathogenic role of decreased Notch signaling in this dystrophy, since only increasing Notch signaling is enough to avoid the phenotype. New data in Figure 8.

More specific comments are as follows:

1. The data from the muscle biopsies do not clearly show a dystrophic pattern as stated in the text.

It is necessary to clarify this point. Patient II.1 is the only one whose biopsy was taken from an affected muscle (quadriceps) showing a clear dystrophic pattern as shown in Fig 1B and supplementary figure S2 (H&E and Gomori Trichrome show increased connective tissue and some necrotic fibers). In contrast, the new biopsies were primarily taken to obtain myoblasts for proliferation and differentiation experiments. Thus, they were taken from mildly affected muscles (biceps brachii) in order to avoid, as much as possible, the effects of the dystrophic process on the results. In the revised version of main Fig 1, we have added a more severe dystrophic image and we have removed the biopsies intentionally taken from the weakly affected muscle, to only show their morphology in supplementary Fig S2.

2. I was unclear as to whether POGLOT1 may act on other proteins with EGF like repeats and thus influence the clinical phenotype.

Rana *et al.* (*O*-Glucose Trisaccharide Is Present at High but Variable Stoichiometry at Multiple Sites on Mouse Notch1, JBC 2011, 286, 31623-37.) listed the proteins predicted to be modified with *O*-glucose. Among them, agrin was previously implicated in a muscle disease. For this reason, we studied agrin in the muscle of our patients by western blot and both the level and molecular weight of agrin were normal (supplementary figure S2.D). Nonetheless, it is very hard to conclusively exclude potential involvement of other proteins that are predicted to be modified by POGLOT1 in this phenotype. We performed rescue experiments to support the involvement of Notch in muscle in flies. We have discussed this issue in the Discussion.

3. The difference between laminin and agrin overlay assays is odd and deserves some discussion as to why this might be so.

The differential binding of laminin and agrin was unexpected to us, too. Although it is known that LARGE-dependent glycosylation of aDG is involved in the binding of both laminin and agrin, the precise glycan structures recognized by these ligands have not been determined. Agrin, but not laminin, showed normal binding to aDG in patients' muscle, suggesting that these ligands recognize slightly different glycan structures. Binding of laminin to aDG may be more sensitive to the extent of glycan polymerization on aDG than that of agrin. Alternatively, glycan structures that are recognized by agrin, such as sialic acids, might not change. It is certainly worth investigating whether the defect of Notch signaling in satellite cells in our patients causes quantitative and/or qualitative change of glycan structures of aDG, but this is beyond our present study. We have discussed this issue in the revised version.

4. Methodology is perhaps not as detailed as it might be in places - i.e. electron microscopy - what resin were the specimens embedded in, fine to say standard methods but there are many and also

there is no reference, the antibody to the core protein from Stephan Kroger - what species is this from - is it a monoclonal - a clone name is mentioned - this would help the data to be compared by the readers with other data obtained in dystroglycanopathy patients.

We have explained in detail all these methodological aspects.

5. Pax7 immunolabelling should always be carried out with a basement membrane marker to ensure the correct location of the staining.

We have repeated Pax7 immunolabelling together with a basement membrane marker (collagen-VI) to ensure Pax7+ cells' location (new data in supplementary Supplementary Figure S9F).

Minor comment:

6. Alpha dystroglycan glycosylation is not decreased during the first stages of muscle regeneration it is at a low level which as the authors subsequently state increases. This is different to a decrease.

We have modified the text to clarify this issue.

In conclusion - the hypotheses underlying this investigation are interesting but the data does not prove or disprove them.

Referee #2 (Comments on Novelty/Model System):

The data full significantly short of supporting the authors conclusions. The manuscript is not suitable for publication.

Referee #2 (Remarks):

In this paper Servian-Morilla and colleagues identified a family of dystrophic patients with a missense mutation in POGLUT1. The authors suggest that lack of POGLUT1 enzymatic activity reduces Notch activity, which results in impaired satellite cell function and reduced α -DAG glycosylation. The paper contains significant amount of data, however in its present form, the evidence presented in that paper do not persuasively support the mechanism proposed by the authors. Especially, the authors do not convincingly demonstrate that Notch down regulation is directly causing the dystrophic phenotype that they observe. Additional experiments are needed to support their claim.

Major comments:

- The authors suggest that Notch is the primary cause of the dystrophic phenotype they observed, although this is only indirectly addressed. Experiment on drosophila partially address that point but

it measures muscle formation, which is quite different than adult muscle. The authors need to perform a Notch overexpression experiment on their myoblasts isolated from the dystrophic patients to measure whether it rescues the satellite cell functions (proliferation, self-renewal, differentiation, fusion)

As mentioned in response to referee 1, in order to demonstrate that reduced activity of POGLUT1 impairs muscle development through decrease in Notch signaling we performed rescue experiments on D233E immortalized myoblasts by lentiviral overexpression of NICD. Increasing Notch signaling fully rescued the proliferation defects in D233E myoblasts, which showed the same proliferation rate as controls. NICD overexpression during differentiation not only rescued the phenotype but also reversed the effect, as myotubes appeared later in D233E culture than in control cells without overexpression of NICD (control-LV-GFP). This result supports the dominant pathogenic role of decreased Notch signaling in this dystrophy, since only increasing Notch signaling is enough to avoid the phenotype. New data in Figure 8.

- Figure 5 shows decreased Notch activity and PAX7-expressing cells in the dystrophic patients. The authors suggest that reduced Notch activity leads to impaired satellite cell function, which ultimately results in dystrophic-like muscle phenotype. However, the opposite could also be true, e.g. the decreased Notch activity could be a consequence of the dystrophic phenotype and not the primary cause. For example, myofiber fragility caused by α -DAG hypoglycosylation could induce muscle degeneration and regeneration, which would lead to satellite cell activation and reduced Notch activity. Alternatively, many proteins other than Notch have EGF-like repeats and could thus be affected by mutation in POGLUT1.

Therefore, this is clearly speculative at this point whether the hypoglycosylation of α -DAG is a consequence of reduced Notch activity. This point needs to be addressed to clearly understand the mechanism. For example, siRNA of POGLUT1 on primary myotubes should help to determine whether POGLUT1 can act directly on α -DAG glycosylation. Alternatively, The authors could measure whether a rescue of Notch in their primary myoblast model affects the level of α -DAG glycosylation. If the mechanism proposed by the authors is correct, Notch overexpression should rescue α -DAG glycosylation, otherwise these 2 processes are parallel, or α -DAG glycosylation acts upstream of Notch.

Indeed, we have thought about the possibility that α DG hypoglycosylation was upstream of Notch. For this reason we collected skeletal muscle from patients suffering from known dystroglycanopathies due to mutations in *POMT2* and *FKTN* genes respectively, which lead to a primary defect of α DG glycosylation (supplementary table S6), to explore Notch and the pool of Pax7+ cells. Despite the dramatic reduction of α DG glycosylation in these muscles, the level of NICD was normal (Supplementary figure S9). In addition, we quantified the number of Pax7+ cells in muscles from dystroglycanopathies and found the same number as controls, which was higher than the number of Pax7+ cells in POGLUT1^{D233E} muscles (Supplementary figure S9). These results support the idea that the reduced Notch activity and reduced number of Pax7+ cells observed in our patients are not consequences of the α DG hypoglycosylation; in other words, the results support that α DG hypoglycosylation in our patients is not acting upstream of Notch.

However, we agree with this reviewer: we did not fully address whether α DG hypoglycosylation is the consequence of the reduced Notch activity or both are parallel phenomena. The first step was to check α DG glycosylation in D233E primary myoblasts during differentiation, and they showed a lower level and irregular progression profile of α DG glycosylation compared to controls, which reproduced the same pattern previously demonstrated in C2C12 cells. Later, we wanted to compare α DG glycosylation by western blot on D233E immortalized myoblasts overexpressing NICD to control-GFP, after five days on differentiation medium. As we have already explained, the NICD overexpression by lentiviral infection increased Notch signaling strongly inhibit the formation of myotubes. Thus, we had to exclude this approach.

To clearly address this point, we have reproduced the experiment performed by Goddeeris et al (Nature 2011) differentiating C2C12 cells during five days and checking aDG glycosylation, but this time we added DAPT (a gamma-secretase inhibitor) to the differentiation medium in order to decrease Notch activation. Under these conditions, we found defective aDG glycosylation during the differentiation, suggesting that proper activation of Notch is required for normal aDG glycosylation during myogenic differentiation. These experiments strongly suggest that Notch reduction is upstream of aDG hypoglycosylation (new data in Figure 9).

- Figure 7 shows impaired myoblast function in the dystrophic patient. However, this experiment was done only on cells from one patient and one control. It is impossible to draw any conclusion from a single biological sample (even if technical replicates were done) because there is multiple other factors that could impact myoblast function other than the mutation in POGlut1, such as age of the patient, gender, intrinsic variations, activation level of satellite cells when the biopsy was performed, etc. This experiment absolutely needs to be performed with biological triplicates both for the controls and the dystrophic patients. This reviewer understand the technical limitation of performing more biopsies on these patients (if it wasn't done previously), however, this figure is submitted to too many bias if it relies only on one patient, and this figure is critical to support the authors conclusions.

As described in response to reviewer 1, we have repeated the experiments adding one more patient (n=2), three healthy controls (n=3) and three dystrophic controls (n=3) (new data in Figure 7). To compensate that we have only n=2 for patients, in addition we have demonstrated the same phenotype using immortalized myoblasts and myogenic converted fibroblasts from patients (n=3) (new data in Supplementary Figure S11 and S12). We think all these approaches strongly demonstrate the phenotype of this disease.

- POGlut1 knockdown on isolated myofibers (from wildtype mice) would improve the author's conclusions regarding the effect of this protein on satellite cell function. The authors could then measure more directly and precisely the effect of POGlut1 on satellite cell proliferation (total number of myogenic cells at 72h), differentiation (number of myogenin-positive cells at 72h), and self-renewal (number of pax7+/MyoD- cells at 72h).

We appreciate the effort of this referee suggesting so many different approaches to strengthen our conclusions. We would like to point out that the patient cells are basically equivalent to POGlut1 knockdown cells, as both reduce activity of POGlut1. The comparison to controls and rescue of proliferation by NICD supports the hypothesis. Accordingly, we were not sure whether using mouse cells to replicate this would add more weight to the study.

- The authors argue that the decrease in Notch signaling and in PAX7-expressing cells in the D233E patients is not secondary to aDG hypoglycosylation by comparing with other types of dystroglycanopathies. However, this might be reflecting the different levels of muscle damage. Only 1 out of 4 of the D233E patients present signs of degeneration (based on H&E staining in Fig 1) while 5 out of 8 disease controls patients have moderate to severe signs of dystrophy (based on suppl table S6). Higher levels of muscle degeneration will obviously lead to higher number of PAX7-positive cells. To draw any conclusion from this, the authors should also compare the patients based on intensity of the disease.

We agree with this referee that higher levels of muscle degeneration will obviously lead to higher number of PAX7+ cells. The dystrophic process activates proliferation of satellite in the first stages but at the end the pool is exhausted. That's why we

compared D233E samples to dystrophic muscles in different stages, to avoid criticisms arguing that the reduction of PAX7+ cells in D233E patients could be a consequence of the dystrophic process. However, as this referee notes, only one D233E sample presents obvious dystrophic features, while the three remaining samples were very mildly affected. As we explained before, we selected muscles that showed normal strength at the moment of biopsy in purpose because our main point was to see if the levels of Notch signaling and the pool of satellite cells were decreased in our patients as a consequence of the nature of the disease (D233E mutation) or was a non-specific consequence of the dystrophic process. For this reason we compared the patients to healthy controls. If the dystrophic process was the cause of the Notch and PAX7+ cells reduction, control muscles should show reduction in the number of Pax7+ cells compared to D233E patients, but the result was the opposite: when biopsies are taken from D233E muscle and healthy controls in similar morphological conditions, the pool of satellite cells in D233E patients show a significant reduction compared to healthy controls, which supports that these findings are related to the nature to the disease.

To fully answer this point we compared the patients and disease controls selecting only mild dystrophies, as this referee suggested, and the results confirm our previous conclusions (new data in Supplementary Figure S9C and D).

Specific points:

- Different samples from different patients were used for the various experiments. For example, patients 2 and 5 were used for figure 2a-b, but patient 3 and 4 were used for figure 2c. It is also unknown which patient was used for figure 7 (if it is patient 1 it could explain that the results are different than control since he has a more severe phenotype and his satellite cells were probably already activated when they were isolated).

We did not repeat a biopsy and use muscle from II.1 patient for culture or Pax7 quantification, in part because of the issue that the reviewer has raised. We were worried that the results might be biased because the biopsy was taken from a severely affected muscle. To avoid the potential confounding effects of using severely affected muscle on the number and biology of myoblasts, the proliferation and differentiation studies were all performed using very mildly affected muscles from patients, in order to be consistent with the results (biopsies obtained from biceps brachii muscles from II.4 and II.5, are very mildly affected as this referee has already observed).

Anyhow, it should be explicitly stated which one, why and how each samples were chosen for each experiments.

We apologize for not specifying the source of each data point in the manuscript. Now, we have clearly explained which samples were used for each experiment modifying the supplementary Table S6. For each experiment we used all the patients' samples available at that moment. We had frozen muscle sample for patient II.1 because it was performed years ago for diagnostic purposes, but he refused a new biopsy. That's why we used this sample only for morphological studies, mRNA and the initial western blot for alpha-dystroglycan, but not for more recent studies. Years later, when we discovered the *POGLUT1* mutation, we asked the patients for a new muscle biopsy. Patients II.2, II.4, II.5 and the healthy sibling II.3 agreed.

- Isolation of satellite cells from human muscle samples could be challenging. Isolation with CD56 microbeads is a relatively good method, although contamination remains a concern. Myoblasts seem to proliferate more slowly in the D233E patient, thus a small fibroblasts contamination could take

over the myoblast population after many days in culture (especially if the fibroblast are not affected by the mutation). Purity of myoblast population needs to be shown.

We have checked the purity by performing immunohistochemistry for desmin in cultures. The purity is over 90% in every culture (new data in the Supplementary Figure S10).

2nd Editorial Decision

30 May 2016

Thank you for the submission of your manuscript to EMBO Molecular Medicine.

We have now heard back from the two Reviewers whom we asked to re-evaluate your manuscript.

You will see that while Reviewer 1 is now supportive, Reviewer 2 is still not satisfied that all issues raised were adequately addressed. Although I will not go into detail, I would like to mention the main points.

The reviewer notes that the important rescue experiment with the NICD vector is lacking a crucial control. S/he also notes that DAPT is not a specific enough Notch inhibitor and is used at a very high concentration, suggesting that a more specific approach should be used. Reviewer 2 also lists other issues, some quite critical, that require attention.

As you know, it is EMBO Molecular Medicine policy to allow a single round of revision only. Nevertheless, we have now re-discussed your manuscript and are prepared in this case to allow you to submit a re-revised version in the light of these comments. We do agree in fact, that the Reviewer's points are well-taken and require direct (and experimental where required) action, which must address each point.

Acceptance or rejection of the manuscript will depend on the completeness of your responses and on the outcome of the required experimentation included in the next, final version of the manuscript.

I also suggest that you carefully adhere to our guidelines for publication in your next version, including our new requirements for supplemental data (<http://embomolmed.embopress.org/authorguide#expandedview>) to speed up the pre-acceptance process in case of a favourable outcome. For instance, supplementary information must be converted into Appendix, with appropriately amended callouts in the manuscript. Also, please provide the source data as separate files for each figure.

I look forward to seeing a revised form of your manuscript as soon as possible.

***** Reviewer's comments *****

Referee #1 (Comments on Novelty/Model System):

Overall I feel that the extra work carried out by the authors justifies publication. The evidence supporting a role for Notch and satellite cell activation in patients with these mutations is reasonable but still not conclusive, this would require a mouse model to further investigate additional pathways. They have answered most issues those not dealt with relate to the inability to identify more families with mutations in this gene from their database, and the availability of additional biopsy material for further experiments. At this stage these are valid reasons. Overall the data is clearly presented and allows for critical analysis by your readership - it is highly likely that additional factors operate in the pathogenesis but this paper presents an interesting pathway with evidence drawn from several different approaches.

Referee #1 (Remarks):

The authors have answered most of my queries and the paper is now much improved.

Referee #2 (Comments on Novelty/Model System):

The paper still needs additional controls and experiments in order to be suitable for publication.

Referee #2 (Remarks):

In this revised version of the manuscript the authors made significant improvement to the manuscript. This reviewer appreciates the increased n size in Fig 7 as well as the addition of the figures 8 and 9 that support the mechanism suggested by the authors. However, these additional controls and experiments are still required for the paper to be suitable for publication.

Major points

- The rescue experiment with the lentiviral NICD vector is a critical experiment for this paper. That experiment lacks an important control, i.e. control patient treated with the LV-NICD-GFP. Without that control it is impossible to know if the vector is really rescuing the defects in the patient's myoblasts or if it is simply stimulating proliferation through another pathway.
- To avoid the differentiation issues in Fig 9 the authors used DAPT to inhibit Notch activity in C2C12 cells. DAPT is a gamma-secretase inhibitor, which will also affect other proteins that are very important for satellite cell/myoblast function, e.g. CD44 (involved in migration and differentiation; Mylona E et al *J Cell physiol* 2006), N-cadherin (myoblast fusion; Mege RM et al *JCS* 1992, Knudsen KA et al *Exp Cell res* 1990), Syndecan-3 (satellite cell activation and proliferation; Cornelison DD et al; *Dev Biol* 2001), and others. This is especially concerning since the authors used DAPT at a very high concentration (50uM). The authors should add a more specific and potent inhibitor such as LY3039478, or any other more specific approach.

Minor points

- The authors mentioned that NICD overexpression reverses the differentiation phenotype. Actually, it looks like it completely suppresses differentiation. It has been shown by many labs that Notch pathway overactivation suppresses myoblast differentiation (Wen Y *Mol Cell Biol* 2012; Brack AS *Cell stem cell* 2008; Kopan R *development* 1994; Jiang C *Dis Model Mech* 2014). The authors should modify their conclusions accordingly.
- In Fig 9A the authors mentioned that the level of glycosylated alpha-Dag is low and irregular, however, it seems similar to control based on the blots shown. Curiously, Fig 9B shows that NICD overexpression reduces Dag glycosylation. The authors mentioned that it is related to the impaired differentiation. Nonetheless, these results are in direct contradiction with their previous observations and their working hypothesis.
- In the introduction the authors state that glycosylation of Dag is not needed for myoblast proliferation. To support that statement they cite 2 papers that analyzed glycosylation in primary myoblasts or C2C12. However, many other papers showed that Dag expression and glycosylation is needed for satellite cell function. Cohn RD et al (*Cell* 2002) first showed that Dag is expressed in satellite cells, where it is needed to promote their regenerative capacity. Dumont NA et al (*Nat Med* 2015) also showed that dystrophin and Dag expression in satellite cells are needed for the generation of differentiated myogenic progenitors and muscle regeneration. Kanagawa M et al (*Hum Mol Ther* 2013) showed that fukutin is needed for myogenic precursor cell proliferation, differentiation and muscle regeneration. Similar results were also observed by Beedle AM et al (*JCI* 2012). Ross J et al (*Stem cells* 2012) showed that glycosylation defect caused by Large deletion impairs satellite cell function, although the authors suggested that it is caused by factors extrinsic to the satellite cells. Technical issues could explain the differences between these papers and the papers cited by the authors, for example it is possible that glycosylation of Dag is needed for satellite cell function in vivo (when satellite cells are in contact with their niche), but not needed for primary myoblast proliferation in vitro when they are cultured in a dish. Anyhow, the authors should modify their introduction accordingly.
- Figure legends should mention the number of representative pictures taken. This is especially

important when the data are not quantified. For example, the authors state in Fig. 2C that there is no ultrastructural changes in muscles but they only show one picture to support that claim.

- The author should discuss the recent paper published in Cell (Vieira NM et al 2015) showing that Jagged1 overexpression mitigates the dystrophic phenotype in golden retriever muscular dystrophic dogs. A process that is, at least, partially mediated by increased myoblast proliferation.

- Pax7 staining on primary myoblasts in Fig. 8A and S11 seems unspecific (faint and partially cytoplasmic) although it looks pretty good in other figures (Fig 7A, Fig S9). How did the author were able to quantify what is positive and what is negative based on the pictures shown in Fig 8A and S11?

2nd Revision - authors' response

16 July 2016

Major points

• The rescue experiment with the lentiviral NICD vector is a critical experiment for this paper. That experiment lacks an important control, i.e. control patient treated with the LV-NICD-GFP. Without that control it is impossible to know if the vector is really rescuing the defects in the patient's myoblasts or if it is simply stimulating proliferation through another pathway.

To address this point, we performed the proliferation assay using immortalized healthy control myoblasts treated with LV-GFP and with LV-NICD-GFP (A, B). As expected, there was no difference in the proliferation rate between the two conditions (see the image below; if the reviewer feels that these data should be included in the manuscript as an Appendix Figure, we would be happy to do so). This result supports the conclusion that reduced Notch signaling is the main reason for the slow proliferation in patients' myoblasts, because NICD overexpression is rescuing the proliferation defect in patient cells but is not stimulating proliferation through any other pathway in controls. Although many labs have previously shown that Notch pathway overactivation suppresses myoblast differentiation (Wen Y Mol Cell Biol 2012; Brack AS Cell stem cell 2008; Kopan R development 1994; Jiang C Dis Model Mech 2014, as the Reviewer also mentioned in Minor point 2), we wanted to repeat the experiment also using both controls (LV-GFP and LV-NICD-GFP). LV-GFP myoblasts showed a regular differentiation process while in LV-NICD-GFP myoblast culture differentiation was suppressed, as expected (C). For reasons that we do not know, NICD cells did not attach to gelatin well, so we cultured them in the absence of gelatin.

- To avoid the differentiation issues in Fig 9 the authors used DAPT to inhibit Notch activity in C2C12 cells. DAPT is a gamma-secretase inhibitor, which will also affect other proteins that are very important for satellite cell/myoblast function, e.g. CD44 (involved in migration and differentiation; Mylona E et al J Cell physiol 2006), N-cadherin (myoblast fusion; Mege RM et al JCS 1992, Knudsen KA et al Exp Cell res 1990), Syndecan-3 (satellite cell activation and proliferation; Cornelison DD et al; Dev Biol 2001), and others. This is especially concerning since the authors used DAPT at a very high concentration (50uM). The authors should add a more specific and potent inhibitor such as LY3039478, or any other more specific approach.

Differentiation of C2C12 cells with medium containing LY3039478 showed a reduced expression of glycosylated aDG at differentiation days 1, 3 and 5 compared to non-treated cells. These findings confirm the previous results obtained with the DAPT treatment. This new data has been added to Figure 9.

Minor points

- The authors mentioned that NICD overexpression reverses the differentiation phenotype. Actually, it looks like it completely suppresses differentiation. It has been shown by many labs that Notch pathway overactivation suppresses myoblast differentiation (Wen Y Mol Cell Biol 2012; Brack AS Cell Stem Cell 2008; Kopan R Development 1994; Jiang C Dis Model Mech 2014). The authors should modify their conclusions accordingly.

We agree with the Reviewer. NICD overexpression reverses the differentiation phenotype so much that it suppresses differentiation. The fact that the outcome of our experiment in patient's myoblasts was the same as what has been shown in control cells supports the notion that reduced Notch signaling is the major contributing factor in stimulating the differentiation of the patients' myoblasts. If pathways other than Notch were responsible for stimulating differentiation as a result of the D233E POGlut1 mutation, NICD overexpression would not have been able to reverse the phenotype by itself, as it does with control cells. We have modified the text to clarify this issue.

- In Fig 9A the authors mentioned that the level of glycosylated alpha-Dag is low and irregular, however, it seems similar to control based on the blots shown. Curiously, Fig 9B shows that NICD overexpression reduces Dag glycosylation. The authors mentioned that it is related to the impaired differentiation. Nonetheless, these results are in direct contradiction with their previous observations and their working hypothesis.

- We think that aDG glycosylation during differentiation in patients is lower than in controls. To avoid subjective interpretations in the blot shown in Fig 9A, we left the non-specific band above the aDG band, which helps to clearly see differences in molecular weight at the end of the differentiation process (D5, last lanes). This result was confirmed using immortalized myoblasts (Fig 9B) where aDG in control-LV-GFP shows a lower molecular weight band at D5 than in patient-LV-GFP.

- We agree with the reviewer that because of its dramatic effect on differentiation, assessing the effect of NICD overexpression on aDG was not informative. In this sense we could remove the panel 9B. However, we think that this panel clearly shows that aDG glycosylation is different between controls and patients at D5.

- In the introduction the authors state that glycosylation of Dag is not needed for myoblast proliferation. To support that statement they cite 2 papers that analyzed glycosylation in primary myoblasts or C2C12. However, many other papers showed that Dag expression and glycosylation is needed for satellite cell function. Cohn RD et al (Cell 2002) first showed that Dag is expressed in satellite cells, where it is needed to promote their regenerative capacity. Dumont NA et al (Nat Med 2015) also showed that dystrophin and Dag expression in satellite cells are needed for the generation of differentiated myogenic progenitors and muscle regeneration. Kanagawa M et al (Hum Mol Ther 2013) showed that fukutin is needed for myogenic precursor cell proliferation, differentiation and

muscle regeneration. Similar results were also observed by Beedle AM et al (JCI 2012). Ross J et al (Stem cells 2012) showed that glycosylation defect caused by Large deletion impairs satellite cell function, although the authors suggested that it is caused by factors extrinsic to the satellite cells. Technical issues could explain the differences between these papers and the papers cited by the authors, for example it is possible that glycosylation of Dag is needed for satellite cell function in vivo (when satellite cells are in contact with their niche), but not needed for primary myoblast proliferation in vitro when they are cultured in a dish. Anyhow, the authors should modify their introduction accordingly.

We have modified the “Introduction” text to clarify this issue.

- Figure legends should mention the number of representative pictures taken. This is especially important when the data are not quantified. For example, the authors state in Fig. 2C that there is no ultrastructural changes in muscles but they only show one picture to support that claim.

We have modified the figure legends to clarify this issue.

- The author should discuss the recent paper published in Cell (Vieira NM et al 2015) showing that Jagged1 overexpression mitigates the dystrophic phenotype in golden retriever muscular dystrophic dogs. A process that is, at least, partially mediated by increased myoblast proliferation.

We have added a brief paragraph about this paper in the “Discussion” section.

- Pax7 staining on primary myoblasts in Fig. 8A and S11 seems unspecific (faint and partially cytoplasmic) although it looks pretty good in other figures (Fig 7A, Fig S9). How did the author were able to quantify what is positive and what is negative based on the pictures shown in Fig 8A and S11?

Below, we show several figures to better illustrate how the quantification of PAX7 signal was performed:

(A) In the primary myoblast culture, the cells showing nuclear PAX7 labeling were considered as PAX7⁺ cells (arrows). These PAX7⁺ cells, when checked in the PAX7/Topro3 merged images, had to show all the PAX7-stained pixels in the nuclear area, to confirm that they really were PAX7⁺ cells. The cells without nuclear PAX7 labeling were considered as PAX7⁻ cells (arrowheads). As Reviewer 2 noticed, the difference between PAX7⁺ and PAX7⁻ cells was very clear in primary myoblast cultures, while the difference was not so evident in immortalized primary myoblasts (B and C). When the distribution of PAX7 was analyzed in immortalized cells (infected and non-infected), we found that PAX7 was concentrated mostly in the nucleus, but it was also located at a lower proportion in the perinuclear and cytoplasmic areas, and these results were the same after checking many different conditions for the immunostaining study. Thus, immortalized primary cultures seem to show a less specific PAX7 labeling than primary cultures, and we do not know the reason. To be as accurate as possible during quantification of PAX7 signal, the cells showing higher PAX7 staining intensity in the nucleus than in the perinuclear space were considered as PAX7⁺ cells (arrows). When nuclear PAX7 stain intensity was equal or lower than perinuclear and cytoplasmic signal, the cells were considered PAX7⁻ (arrowheads).

A. Primary myoblasts culture

B. Immortalized primary myoblasts culture

C. Immortalized primary myoblasts culture infected with lentivirus

3rd Editorial Decision

08 August 2016

Thank you for the submission of your revised manuscript to EMBO Molecular Medicine. We have now received the enclosed report from Reviewer 2, who was asked to re-assess it. As you will see s/he is now globally supportive and I am pleased to inform you that we will be able to accept your manuscript pending the following final amendments:

- 1) Please comply with the reviewer's final request to include the additional control (healthy control myoblasts treated with LVGFP and with LVNICDGFP) that you show in the rebuttal letter in the appendix. Please make sure you add the relative callout in the appropriate place in the manuscript.
- 2) Please add a table of contents on the first page of the appendix file as per our Author Guidelines.

- 3) Please correct "Appendix Fig 5A" on page 8 to "Appendix Fig S5A"
- 4) Please note that Appendix Figure S12 and its legend need to be corrected as there are 4 panels marked A, B, G, H and I, with no C and D...
- 5) The source data labeled Fig. 9E should perhaps be re-labelled to Fig. 9D
- 6) Please provide the source data tables for the graphs as Excel or CVS files. See also point 7 below.
- 7) Please present the source data as ONE separate file per figure (even if the figure is composed of multiple panels), as per our Author Guidelines.
- 8) As per our Author Guidelines, the description of all reported data that includes statistical testing must state the name of the statistical test used to generate error bars and P values, the number (n) of independent experiments underlying each data point (not replicate measures of one sample), and the actual P value for each test (not merely 'significant' or 'P < 0.05').
- 9) Every published paper now includes a 'Synopsis' to further enhance discoverability. Synopses are displayed on the journal webpage and are freely accessible to all readers. They include a short standfirst as well as 2-5 one sentence bullet points that summarise the paper. Please provide the synopsis including the short list of bullet points that summarise the key NEW findings. The bullet points should be designed to be complementary to the abstract - i.e. not repeat the same text. We encourage inclusion of key acronyms and quantitative information. Please use the passive voice. Please attach this information in a separate file or send them by email, we will incorporate it accordingly. You are also welcome to suggest a striking image or visual abstract to illustrate your article. If you do please provide a jpeg file 550 px-wide x 400-px high.

Please submit your revised manuscript within two weeks. I look forward to seeing a revised form of your manuscript as soon as possible.

***** Reviewer's comments *****

Referee #2 (Remarks):

The paper is acceptable for publication. The additional control (healthy control myoblasts treated with LVGFP and with LVNICDGFP) that they showed in the rebuttal letter should be included in the supplemental info.

3rd Revision - authors' response

22 August 2016

Regarding the final amendments:

1) Please comply with the reviewer's final request to include the additional control (healthy control myoblasts treated with LVGFP and with LVNICDGFP) that you show in the rebuttal letter in the appendix. Please make sure you add the relative callout in the appropriate place in the manuscript:

Appendix Figure has been added and the relative callout in the manuscript as well

2) Please add a table of contents on the first page of the appendix file as per our Author Guidelines.

Table of Contents has been added on the Appendix

3) Please correct "Appendix Fig 5A" on page 8 to "Appendix Fig S5A"

It has been corrected

4) Please note that Appendix Figure S12 and its legend need to be corrected as there are 4 panels marked A, B, G, H and I, with no C and D...

They have been corrected

5) The source data labeled Fig. 9E should perhaps be re-labelled to Fig. 9D

6) Please provide the source data tables for the graphs as Excel or CVS files. See also point 7 below.

7) Please present the source data as ONE separate file per figure (even if the figure is composed of multiple panels), as per our Author Guidelines.

Source Data have been corrected according the Author Guidelines

8) As per our Author Guidelines, the description of all reported data that includes statistical testing must state the name of the statistical test used to generate error bars and P values, the number (n) of independent experiments underlying each data point (not replicate measures of one sample), and the actual P value for each test (not merely 'significant' or 'P < 0.05').

We have included exact P value for each test

9) Every published paper now includes a 'Synopsis' to further enhance discoverability. Synopses are displayed on the journal webpage and are freely accessible to all readers. They include a short standfirst as well as 2-5 one sentence bullet points that summarise the paper. Please provide the synopsis including the short list of bullet points that summarise the key NEW findings. The bullet points should be designed to be complementary to the abstract - i.e. not repeat the same text. We encourage inclusion of key acronyms and quantitative information. Please use the passive voice. Please attach this information in a separate file or send them by email, we will incorporate it accordingly. You are also welcome to suggest a striking image or visual abstract to illustrate your article. If you do please provide a jpeg file 550 px-wide x 400-px high.

Synopsis text and Synopsis Figure (.JPEG file) have been uploaded as "Related Manuscript File"

Please, do not hesitate to contact us for further details.

Corresponding Author Name: Carmen Paradás

Manuscript Number: 2015-05815-V2